# SKILL MACHINES: TEMPORAL LOGIC COMPOSITION IN REINFORCEMENT LEARNING

## ABSTRACT

A major challenge in reinforcement learning is specifying tasks in a manner that is both interpretable and verifiable. One common approach is to specify tasks through reward machines—finite state machines that encode the task to be solved. We introduce skill machines, a representation that can be learned directly from these reward machines that encode the solution to such tasks. We propose a framework where an agent first learns a set of base skills in a reward-free setting, and then combines these skills with the learned skill machine to produce composite behaviours specified by any regular language and even linear temporal logics. This provides the agent with the ability to map from complex logical task specifications to near-optimal behaviours zero-shot. We demonstrate our approach in both a tabular and high-dimensional video game environment, where an agent is faced with several of these complex, long-horizon tasks. Our results indicate that the agent is capable of satisfying extremely complex task specifications, producing near optimal performance with no further learning. Finally, we demonstrate that the performance of skill machines can be improved with regular off-policy reinforcement learning algorithms when optimal behaviours are desired.

## 1 INTRODUCTION

Reinforcement learning (RL) is a promising framework for developing truly general agents capable of acting autonomously in the real world. Despite recent successes in the field, ranging from video games (Badia et al., 2020) to robotics (Levine et al., 2016), there are several shortcomings to existing approaches that hinder RL's real-world applicability. One issue is that of sample efficiency—while it is possible to collect millions of data points in a simulated environment, it is simply not feasible to do so in the real world. This inefficiency is exacerbated when a single agent is required to solve multiple tasks (as we would expect of a generally intelligent agent). One approach of generally intelligent agents to overcoming this challenge is their ability to reuse learned behaviours to solve new tasks (Taylor & Stone, 2009), preferably without further learning. That is, to rely on *composition*, where an agent first learns individual skills and then combines them to produce novel behaviours. There are several notions of compositionality in the literature, such as temporal composition, where skills are invoked one after the other ("pickup a blue object then a box") (Sutton et al., 1999; Barreto et al., 2019), and spatial composition, where skills are combined to produce a new behaviour to be executed ("pickup a blue box") (Todorov, 2009; Saxe et al., 2017; Van Niekerk et al., 2019; Alver & Precup, 2022).

Notably, work by Nangue Tasse et al. (2020) has demonstrated how an agent can learn skills that can be combined using Boolean operators, such as negation and conjunction, to produce semantically meaningful behaviours without further learning. An important, additional benefit of this compositional approach is that it provides a way to address another key issue with RL: tasks, as defined by reward functions, can be notoriously difficult to specify. This may lead to undesired behaviours that are not easily interpretable and verifiable. Composition that enables simpler task specifications and produces reliable behaviours thus represents a major step towards safe AI (Cohen et al., 2021).

Unfortunately, these compositions are strictly spatial. Thus, another issue arises when an agent is required to solve a long horizon task. In this case, it is often near impossible for the agent to solve the task, regardless of how much data it collects, since the sequence of actions to execute before a learning signal is received is too large (Arjona-Medina et al., 2019). This can be mitigated by leveraging higher-order skills, which shorten the planning horizon (Sutton et al., 1999). One specific implementation

of this is *reward machines*—finite state machines that encode the tasks to solve (Icarte et al., 2018). While reward machines obviate the sparse reward problem, used in isolation, they still require the agent to learn how to solve a given task through environment interaction, and the subsequent solution is monolithic, resulting in the afore mentioned problems with applicability to new tasks and reliability.

In this work, we combine these two approaches to develop an agent capable of zero-shot spatial *and* temporal composition. We particularly focus on temporal logic composition, such as linear temporal logic (LTL) (Pnueli, 1977), allowing agents to sequentially chain and order their skills while ensuring certain conditions are always or never met. We make the following contributions: (a) we propose *skill machines*, a finite state machine that can be autonomously learned by a compositional agent, and which can be used to solve any task expressible as a finite state machine without further learning; (b) we prove that these skill machines are *satisficing*—given a task specification, an agent can successfully solve it while adhering to any constraints; and (c) we demonstrate our approach in several environments, including a high-dimensional video game domain. Having learned a set of base skills in a reward-free setting (in the absence of task rewards from a reward machine), our results indicate that our method is capable of producing near-optimal behaviour for a variety of long-horizon tasks without further learning.

To describe our approach to temporal composition, we use the *Office Gridworld* (Icarte et al., 2018) as a running example. In the environment, illustrated by Figure 1a, an agent (blue circle) can move to adjacent cells in any of the cardinal directions. It can also pick up coffee or mail at locations ☕ or ✉ respectively, and it can deliver them to the office at location 👤. Cells marked ✲ indicate decorations that are broken if the agent collides with them, and cells marked $A$–$D$ indicate the centres of the corner rooms. The reward machines that specify tasks in this environment are defined over 10 propositions: $\mathcal{P} = \{A, B, C, D, ✲, ☕, ✉, 👤, ✉^+, 👤^+\}$, where the first 8 propositions are true when the agent is at their respective locations, $✉^+$ is true when the agent is at $✉$ and there is mail to be collected, and $👤^+$ is true when the agent is at $👤$ and there is someone in the office.

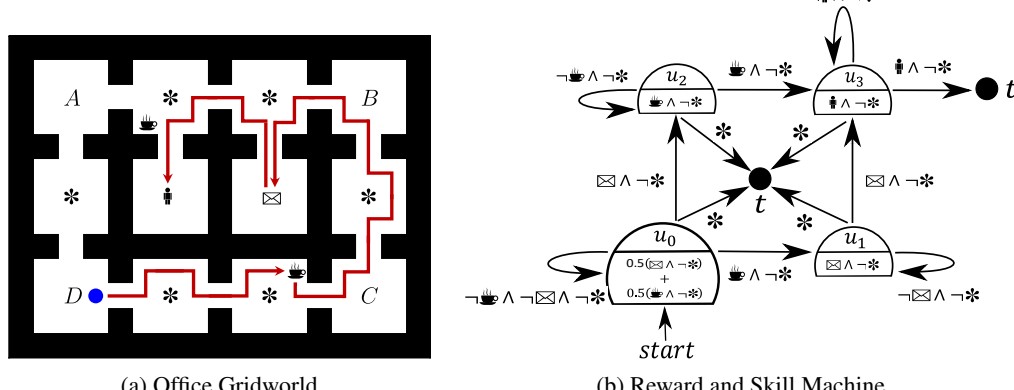

| (a) Office Gridworld | (b) Reward and Skill Machine |
|---|---|

Figure 1: Illustration of (a) the office gridworld where the blue circle represents the agent and (b) the finite state machine representing both the reward and skill machine for the task "deliver coffee and mail to the office without breaking any decoration" where the black dots labeled $t$ represent terminal states. The reward machine gives rewards ($\delta_r$) to the agent for each FSM state and the skill machine gives the composed skills $\delta_Q$ that maximises those rewards. For example at $u_0$, $\delta_r(u_0) = 0.5(R_{✉ \wedge \neg ✲}) + 0.5(R_{☕ \wedge \neg ✲})$ and $\delta_Q(u_0) = 0.5(Q_✉ \wedge \neg Q_✲) + 0.5(Q_☕ \wedge \neg Q_✲)$.

## 2 BACKGROUND

We model the agent's interaction with the world as a Markov Decision Process (MDP), given by $(\mathcal{S}, \mathcal{A}, P, R, \gamma)$, where (i) $\mathcal{S} \subseteq \mathbb{R}^n$ is the $n$-dimensional state space; (ii) $\mathcal{A}$ is the set of (possibly continuous) actions available to the agent; (iii) $P(s'|s, a)$ is the dynamics of the world, representing the probability of the agent reaching state $s'$ after executing action $a$ in state $s$; (iv) $R$ is a reward function bounded by $[R_{\text{MIN}}, R_{\text{MAX}}]$ that represents the task the agent needs to solve; and (v) $\gamma \in [0, 1]$ is a discount factor. The aim of the agent is to compute a Markov policy $\pi$ from $\mathcal{S}$ to $\mathcal{A}$ that optimally solves a given task. Instead of directly learning a policy, an agent will often

instead learn a value function that represents the expected return following policy $\pi$ from state $s$: $V^\pi(s) = \mathbb{E}^\pi \left[ \sum_{t=0}^\infty \gamma^t R(s_t, a_t) \right]$. A more useful form of value function is the action-value function $Q^\pi(s, a)$, which represents the expected return obtained by executing $a$ from $s$, and then following $\pi$. The optimal action-value function is given by $Q^*(s, a) = \max_\pi Q^\pi(s, a)$ for all states $s$ and actions $a$, and the optimal policy follows by acting greedily with respect to $Q^*$ at each state.

## 2.1 Logical Composition in the Multitask Setting

We are interested in the multitask setting, where an agent is required to reach a set of goals in some goal space $\mathcal{G} \subseteq \mathcal{S}$. We assume that all tasks share the same state space, action space and dynamics, but differ in their reward functions. We model this setting by defining a background MDP $M = \langle \mathcal{S}, \mathcal{A}, P, R, \gamma \rangle$ with its own state space, action space, transition dynamics and background reward function: $R$. Any individual task $\tau$ is then specified by a task-specific reward function $R_\tau$ that is non-zero only for states in $\mathcal{G}$. The reward function for the resulting task MDP is then simply $R + R_\tau$.

Nangue Tasse et al. (2020) consider the case where $R_\tau(g, a) \in \{R_{\text{MIN}}, R_{\text{MAX}}\}$ and develop a framework that allows agents to apply the Boolean operations of conjunction ($\wedge$), disjunction ($\vee$) and negation ($\neg$) over the space of tasks and value functions. This is achieved by first defining the goal-oriented reward function $\bar{R}$ which extends the task rewards ($R + R_\tau$) to penalise an agent for achieving goals different from the one it wished to achieve:

$$\bar{R}(s, g, a) := \begin{cases} R_{\text{MISS}} & \text{if } g \neq s; \text{ where } g \in \mathcal{G} \text{ and } s \text{ is absorbing} \\ R(s, a) + R_\tau(s, a) & \text{otherwise,} \end{cases} \tag{1}$$

$R_{\text{MISS}}$ is a large negative penalty that can be derived from the bounds of the reward function. Using Equation 1, we can define the related goal-oriented value function as:

$$\bar{Q}(s, g, a) = \bar{R}(s, g, a) + \gamma \int_{\mathcal{S}} \bar{V}^{\bar{\pi}}(s', g) P_{(s,a)}(ds'), \tag{2}$$

where $\bar{V}^{\bar{\pi}}(s, g) = \mathbb{E}_{\bar{\pi}} \left[ \sum_{t=0}^\infty \gamma^t \bar{R}(s_t, g, a_t) \right]$.

If a new task can be represented as the logical expression of previously learned tasks, Nangue Tasse et al. (2020) prove that the optimal policy can immediately be obtained by composing the learned goal-oriented value functions using the same expression. For example, the union ($\vee$), intersection ($\wedge$), and negation ($\neg$) of two goal-reaching tasks $A$ and $B$ can be solved as follows (we omit the value functions' parameters for readability):

$$\bar{Q}^*_{A \vee B} = \bar{Q}^*_A \vee \bar{Q}^*_B := \max\{\bar{Q}^*_A, \bar{Q}^*_B\}; \qquad \bar{Q}^*_{A \wedge B} = \bar{Q}^*_A \wedge \bar{Q}^*_B := \min\{\bar{Q}^*_A, \bar{Q}^*_B\};$$

$$\bar{Q}^*_{\neg A} = \neg \bar{Q}^*_A := (\bar{Q}^*_{SUP} + \bar{Q}^*_{INF}) - \bar{Q}^*_A$$

where $\bar{Q}^*_{SUP}$ and $\bar{Q}^*_{INF}$ are the goal-oriented value functions for the maximum task ($R_\tau = R_{\text{MAX}}$ for all $\mathcal{G}$) and minimum task ($R_\tau = R_{\text{MIN}}$ for all $\mathcal{G}$), respectively. Following Nangue Tasse et al. (2022), we will also refer to these goal-oriented value functions as *world value functions* (WVFs).

## 2.2 Reward Machines

One difficulty with the standard MDP formulation is that the agent is often required to solve a complex long-horizon task using only a scalar reward signal as feedback from which to learn. To overcome this, Icarte et al. (2018) propose *reward machines* (RMs), which provide structured feedback to the agent in the form of a finite state machine (FSM). RMs encode a reward function using a set of propositional symbols $\mathcal{P}$ that represent abstract environment features as follows:

**Definition 1** (Reward Machine). *Given a set of states $\mathcal{S}$ and actions $\mathcal{A}$, a reward machine is a tuple $R_{\mathcal{S}\mathcal{A}} = \langle U, u_0, F, \delta_u, \delta_r \rangle$ where (i) $U$ is a finite set of states; (ii) $u_0 \in U$ is an initial state; (iii) $F$ is a finite set of terminal states; (iv) $\delta_u : U \times [S \times A \times S] \to U \cup F$ is the state-transition function; and (v) $\delta_r : U \to [\mathcal{S} \times \mathcal{A} \times \mathcal{S} \to \mathbb{R}]$ is the state-reward function.*

RMs consist of a finite set of states $U$ where transitions between RM states are governed by $\delta_u$, and where each RM state emits a reward function according to $\delta_r$. To incorporate RMs into the RL framework, the agent must be able to determine a correspondence between abstract RM propositions

($\mathcal{P} = \{A, B, C, D, \maltese, \text{✋}, \boxtimes, \text{♟}, \boxtimes^{+}, \text{♟}^{+}\}$ for example) and states in the environment. To achieve this, the agent is equipped with a labelling function $L : \mathcal{S} \times \mathcal{A} \times \mathcal{S} \to 2^{\mathcal{P}}$ that assigns truth values to the propositions based on the agent's interaction with its environment. Thus, $2^{\mathcal{P}} \subset S \times A \times S$ depicts an equivalence class from $S \times A \times S$. A particular instantiation of an RM that is used in practice—for example when converting an LTL specification to an RM—is a *simple reward machine* (SRM denoted similarly as $R_{\mathcal{P}\mathcal{A}}$), which restricts the form of the state-reward function to be $\delta_r : U \to [2^{\mathcal{P}} \to \mathbb{R}]$ (Icarte et al., 2018). In other words, the SRM state-reward function emits a function which maps the simpler equivalence class of states to a scalar reward.

The agent's aim then is to learn a policy $\pi : \mathcal{S} \times U \to \mathcal{A}$ over the joint background MDP and RM (MDPRM), which is defined by the tuple $\mathcal{T} = \langle \mathcal{S}, \mathcal{A}, P, \gamma, \mathcal{P}, L, U, u_0, F, \delta_u, \delta_r \rangle$. However, the rewards from the reward machine are not necessarily Markov with respect to the environment. Icarte et al. (2018) shows that a **product MDPRM** (Definition 2 below) guarantees that the rewards are Markov such that the policy can be learned with standard algorithms like $Q$-learning(Icarte et al., 2018). This is because the product MDPRM uses the cross-product to consolidate how actions in the environment result in simultaneous transitions in the environment and state machine. Thus, product MDPRMs take the form of standard, learnable MDPs.

**Definition 2** (Product MDPRM). *Let $\mathcal{T} = \langle \mathcal{S}, \mathcal{A}, P, \gamma, \mathcal{P}, L, U, u_0, F, \delta_u, \delta_r \rangle$ be an MDPRM. The product MDPRM is then defined by the tuple $M_{\mathcal{T}} = \langle \mathcal{S}_{\mathcal{T}}, \mathcal{A}, P_{\mathcal{T}}, R_{\mathcal{T}}, \gamma \rangle$ where $\mathcal{S}_{\mathcal{T}} \coloneqq \mathcal{S} \times (U \cup F)$, $R_{\mathcal{T}}(\langle s, u \rangle, a, \langle s', u' \rangle) \coloneqq \delta_r(u)(s, a, s')$,*

$$P_{\mathcal{T}}(\langle s, u \rangle, a) \coloneqq \begin{cases} \langle s', u' \rangle & \textit{if } u \in U \\ \langle s', u \rangle & \textit{otherwise} \end{cases}, s' \sim P(\cdot | s, a) \textit{ and } u' = \delta_u(u, (s, a, s')).$$

## 3    LEVERAGING SKILL COMPOSITION FOR TEMPORAL LOGIC TASKS

Since we are interested in temporal logic tasks, we will restrict our attention to RMs whose rewards per node are specified by linear preferences over Boolean expressions (instead of arbitrary real-valued functions that are not grounded in achieving goals in an environment):

**Definition 3** (Tasks). *Let $M = \langle \mathcal{S}, \mathcal{A}, P, R, \gamma \rangle$ be a background MDP. A task is a product MDPRM $M_{\mathcal{T}} = \langle \mathcal{S}_{\mathcal{T}}, \mathcal{A}, P_{\mathcal{T}}, R_{\mathcal{T}}, \gamma \rangle$ over $M$ and a reward machine with reward function*

$$\delta_r(u) \in \left\{ R_{\mathbf{w}}(s, a, s') = \sum_{p \in 2^{2^{\mathcal{P}}}} w_p R_p(s, a, s') : R_p(s, a, s') \sum_{p \in 2^{2^{\mathcal{P}}}} w_p = 1 \textit{ and } \mathbf{w} \in \mathbb{R}^{\left| 2^{2^{\mathcal{P}}} \right|} \right\}, \textit{ where}$$

$$R_p(s, a, s') \coloneqq \begin{cases} R_{MAX} & \textit{if } L(s, a, s') \in p \\ R_{MIN} & \textit{if } L(s, a, s') \notin p \\ R(s, a) & \textit{otherwise.} \end{cases}$$

We will assume that the rewards $R_p(s, a, s')$ are such that the policies that maximises them are guaranteed to reach states where the corresponding propositions $p$ are true—a common example is to have $R(s, a) = R_{\text{MIN}} = 0$ and $R_{\text{MAX}} = 1$. This definition of RMs provides a general notion of tasks that are still grounded in achieving goals. Figure 1b illustrates an example of an RM in the office gridworld for solving the task "deliver coffee and mail to the office without breaking any decoration".

### 3.1    FROM ENVIRONMENT TO PRIMITIVES

In order to solve temporal logic tasks zero-shot, we propose to first learn a set of primitive skills which can later be composed to maxise the rewards per RM node without further learning. To achieve this, we first introduce the concept of *constraints* $\mathcal{C} \subseteq \mathcal{P}$, which are the set of propositions that an agent should avoid setting to true and corresponds to the global operator $G$ in a linear temporal logic (LTL) specification. An example of a constraint might be that the agent should complete a task, but avoid breaking any decorations while doing so ($\mathcal{C} = \{\maltese\}$ and in the LTL we say $G \neg \maltese$). We can now define the notions of task primitives and skill primitives such as "Pick up coffee" ($F \text{✋}$ in LTL) or "don't break any decoration" ($\neg(F\maltese) = G \neg\maltese$ in LTL).

**Definition 4** (Primitives). *Let $M = \langle \mathcal{S}, \mathcal{A}, P, R, \gamma \rangle$ be a background MDP. We define a task primitive in this domain as $M_p = \langle \mathcal{S}_{\mathcal{G}}, \mathcal{A}_{\mathcal{G}}, P_{\mathcal{G}}, R_p, \gamma \rangle$, $p \in 2^{2^{\mathcal{P}}}$, with absorbing goal space $\mathcal{G} = 2^{\mathcal{P}}$ and labelling function $L$, where*

$$\mathcal{S}_{\mathcal{G}} \coloneqq (\mathcal{S} \times 2^{\mathcal{C}}) \cup 2^{\mathcal{P}}, \text{where } \mathcal{C} \text{ is the set of constraints;}$$

$$\mathcal{A}_{\mathcal{G}} \coloneqq \mathcal{A} \times \mathcal{A}_{\tau}, \text{ where } \mathcal{A}_{\tau} = \{0, 1\} \text{ represents whether or not to terminate a task;}$$

$$P_{\mathcal{G}}(\langle s, c \rangle, \langle a, a_{\tau} \rangle) \coloneqq \begin{cases} L(s, a, s') & \text{if } a_{\tau} = 1 \\ \langle s', c' \rangle & \text{otherwise} \end{cases},$$

$$\text{where } s' \sim P(\cdot | s, a) \text{ and } c' = c \cup (\mathcal{C} \cap L(s, a, s'));$$

$$R_p(\langle s, c \rangle, \langle a, a_{\tau} \rangle) \coloneqq \begin{cases} R_{MAX} & \text{if } a_{\tau} = 1 \text{ and } L(s, a, s') \in p \\ R_{MIN} & \text{if } a_{\tau} = 1 \text{ and } L(s, a, s') \notin p \\ R(s, a) & \text{otherwise.} \end{cases}$$

*A skill primitive $\bar{Q}_p^*$ is defined as the WVF for the task primitive $M_p$.*

The above defines the state space of primitives to be the product of the environment states and the set of constraints, incorporating the set of propositions that are currently true. The action space is augmented with a terminating action following Barreto et al. (2019) and Nangue Tasse et al. (2020), which indicates that the agent wishes to achieve the goal it is currently at, and is similar to an option's termination condition (Sutton et al., 1999). The transition dynamics update the environment state and constraints set to true when a regular action is taken, and use the labelling function to return the set of propositions achieved when the agent decides to terminate. Finally, the agent receives the regular background reward when taking an action, but a primitive-specific goal reward when it terminates.

Importantly, primitives are *temporally atomic*, that is, they correspond to tasks with a single non-terminal RM state. They are, thus, the smallest unit of temporal logic. However, since the goal space of task primitives are defined by Boolean propositions, we can leverage prior work to solve any logical composition over them by composing their corresponding skill primitives (Nangue Tasse et al., 2020). We will denote the set of base task primitives to be $\mathcal{M}_{\mathcal{P}}$ and the corresponding base skill primitives $\bar{\mathcal{Q}}_{\mathcal{P}}^*$, which can be composed to obtain any other primitive. For example: "Pick up coffee without breaking any decoration" $((F\text{☕}) \wedge \neg(F\text{❋}))$ is another primitive by Definition 4. As we discuss in Section 3.2, this solves the primary problem with Reward Machines - they suffer from the curse of dimensionality when all possible primitives must be relearned at all states in the FSM. Skill Machines in contrast leverage primitive composition within **and** across FSM states. Theorem 1 below demonstrates that a linear combination of skill primitives maximise the task (in terms of Definition 3) rewards per RM node without further learning (proofs of all theorems are presented in the Appendix). This is also demonstrated experimentally in Figure 8 in Appendix A.5.

**Theorem 1.** *Let $\mathbf{R}_{\mathcal{G}}$ be a vector of rewards for each task primitive, and $\bar{\mathbf{Q}}_{\mathcal{G}}^*$ be the corresponding vector of optimal WVFs. Then, for an MDP $m = \langle \mathcal{S}_{\mathcal{G}}, \mathcal{A}_{\mathcal{G}}, P_{\mathcal{G}}, R_{\mathbf{w}}, \gamma \rangle$ with linear preference reward function $R_{\mathbf{w}} = \mathbf{w} \cdot \mathbf{R}_{\mathcal{G}}$, we have $\bar{Q}_m^* = \mathbf{w} \cdot \bar{\mathbf{Q}}_{\mathcal{G}}^*$.*

## 3.2 FROM TASKS TO SKILL MACHINES

We now have agents capable of solving any logical and linear composition of base task primitives $\mathcal{M}_{\mathcal{P}}$ by only learning their corresponding base skill primitives $\bar{\mathcal{Q}}_{\mathcal{P}}^*$. Given this compositional ability over skills, and reward machines that expose the structure of tasks, agents can solve temporally extended tasks with little or no further learning. To achieve this, we define a skill machine (SM) as a representation of logical and temporal knowledge over skills.

**Definition 5** (Skill Machine). *Given a task $M_{\mathcal{T}} = \langle \mathcal{S}_{\mathcal{T}}, \mathcal{A}, P_{\mathcal{T}}, R_{\mathcal{T}}, \gamma \rangle$ defined by a reward machine $R_{\mathcal{SA}} = \langle U, u_0, F, \delta_u, \delta_r \rangle$, a set of propositional symbols $\mathcal{P}$ with constraints $\mathcal{C} \subseteq \mathcal{P}$, and their corresponding base skill primitives $\bar{\mathcal{Q}}_{\mathcal{P}}^*$, a skill machine is a tuple $\bar{\mathcal{Q}}_{\mathcal{SA}}^* = \langle U, u_0, F, \delta_u, \delta_Q, \mathbf{w}_U, \mathbf{w}_{\mathcal{G}} \rangle$ where (i) $\mathbf{w}_U : U \times U \to \mathbb{R}$ is a preference function over transitions; (ii) $\mathbf{w}_{\mathcal{G}} : \mathcal{S}_{\mathcal{G}} \times \mathcal{G} \to \mathbb{R}$ is a preference function over goals; and (iii) $\delta_Q : U \to [\mathcal{S}_{\mathcal{G}} \times \mathcal{A}_{\mathcal{G}} \to \mathbb{R}]$ is the state-skill function defined by:*

$$\delta_Q(u)(\langle s, c \rangle, \langle a, 0 \rangle) \mapsto \sum_{g \in \mathcal{G}} \sum_{u' \in U} \mathbf{w}_{\mathcal{G}}(\langle s, c \rangle, u, g) \mathbf{w}_U(u, u') \bar{Q}_{u,u'}^*(\langle s, c \rangle, g, \langle a, 0 \rangle),$$

*where $\bar{Q}_{u,u'}^*$ is the WVF obtained by composing the skill primitives $\bar{\mathcal{Q}}_{\mathcal{G}}^*$ according to the Boolean expression for the transition $\delta_u(u)(s, a, s') = u'$.*

For a given state $s \in \mathcal{S}$ in the environment, true constraints $c \in \mathcal{C}$, and state $u$ in the skill machine, the skill machine uses its preference over transitions $\mathbf{w}_U$ and goals $\mathbf{w}_{\mathcal{G}}$ to compute a skill

$Q(\langle s,c \rangle, \langle a,0 \rangle) \coloneqq \delta_Q(u)(\langle s,c \rangle, \langle a,0 \rangle)$ that an agent can use to take an action $a$. The environment then transitions to the next state $s'$ where $\langle s',c' \rangle \leftarrow P_{\mathcal{G}}(\langle s,c \rangle, \langle a,0 \rangle)$ and the skill machine transitions to $u' \leftarrow \delta_u(u, L(s,a,s'))$. $\mathbf{w}_U$ represents cases where there is not necessarily a single desirable transition to follow given the current SM state. This is illustrated by the SM in Figure 1b, where mail (⊠) and coffee (☕) are equally desirable at the initial state. Similarly, $\mathbf{w}_{\mathcal{G}}$ represents cases where there may be a single desirable task, but its goals are not necessarily equally desirable given the environment state—for example when the agent needs to first pick up coffee but there are two coffee locations. Remarkably, there always exists a choice for $\mathbf{w}_U$ and $\mathbf{w}_{\mathcal{G}}$ that is optimal with respect to the corresponding reward machine, as shown in Theorem 2:

**Theorem 2.** *Let $\pi^*(s,u)$ be the optimal policy for a task $M_{\mathcal{T}}$, and let $\mathcal{C} = \mathcal{P}$. Then there exists a corresponding skill machine with a $\mathbf{w}_{\mathcal{G}}$ and $\mathbf{w}_U$ such that $\pi^*(s,u) \in \arg\max_{a \in \mathcal{A}} \delta_Q(u)(\langle s,c \rangle, \langle a,0 \rangle)$, where $\delta_Q$ is given by $\mathbf{w}_{\mathcal{G}}$ and $\mathbf{w}_U$ as per Definition 5.*

Theorem 2 shows that skill machines can be used to solve tasks without having to relearn action level policies. The next section shows how an agent can approximate a skill machine by planning over simple reward machines.

### 3.3 FROM SIMPLE REWARD MACHINES TO SKILL MACHINES

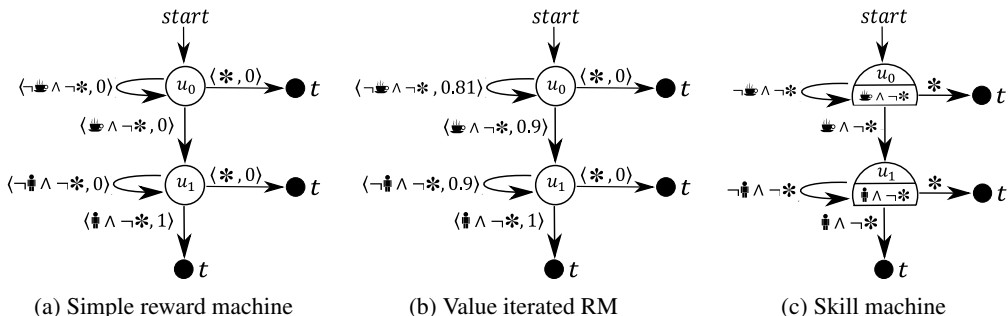

(a) Simple reward machine  (b) Value iterated RM  (c) Skill machine

Figure 2: The SRM, value iterated RM and skill machine for the task "Deliver coffee to the office without breaking any decoration". This task is specified using LTL as $(F(☕ \wedge X(F\,🖥)) ) \wedge (G\,\neg✽))$, where $F = Finally, X = neXt, G = Globally$ are LTL operators. The corresponding RM is obtained by converting the LTL into a finite state machine (Duret-Lutz et al., 2016) and then giving a reward of 1 for accepting transitions and 0 otherwise. The black dots labeled $t$ represent terminal states.

In the previous section, we introduced skill machines and showed that they can be used to represent the logical and temporal composition of skills needed to solve reward machines. We now show how for simple RMs (RMs returning scalar rewards as defined in Section 2.2) their approximate SM can be obtained zero-shot without further learning. To achieve this, we first plan over the reward machine (using value iteration, for example) to obtain Q-values for each transition. We then select the skills for each SM state greedily. This process is illustrated in Figure 2. While this only holds for cases where the greedy skills are always satisfice-able from any environment state, this still covers many tasks of interest. In particular, this holds for any RM with non-zero rewards of $R_{\text{MAX}}$ only at accepting transitions,[1] as shown in Theorem 3.

**Theorem 3.** *Let $R_{\mathcal{P}\mathcal{A}} = \langle U, u_0, F, \delta_u, \delta_{\mathcal{M}} \rangle$ be a satisfice-able simple reward machine with non-zero rewards $R_{MAX}$ only for accepting transitions, and for which all valid transitions $(u,u')$ are achievable from any state $s \in \mathcal{S}$. Define the skill machine $\bar{\mathcal{Q}}^*_{\mathcal{S}\mathcal{A}} = \langle U, u_0, F, \delta_u, \delta_Q, \mathbf{w}_U, \mathbf{w}_{\mathcal{G}} \rangle$ with*

$$\mathbf{w}_U(u,u') \coloneqq 1 \quad \text{if } u' = \arg\max_{u''} Q^*(u,u''), \quad 0 \quad \text{otherwise}$$

$$\mathbf{w}_{\mathcal{G}}(\langle s,c \rangle, u, g) \coloneqq 1 \quad \text{if } g = \arg\max_{g'}\max_a \sum_{u'} \mathbf{w}_U(u,u')\bar{Q}^*_{u,u'}(\langle s,c \rangle, g', \langle a,0 \rangle), \quad 0 \quad \text{otherwise}$$

*where $Q^*$ is the optimal transition-value function for $R_{\mathcal{P}\mathcal{A}}$. Then following the policy $\pi(s,u) \in \arg\max_{a \in \mathcal{A}} \delta_Q(u)(\langle s,c \rangle, \langle a,0 \rangle)$, will reach an accepting transition.*

---

[1]Accepting transitions are transitions at which the high level task—described, for example, by linear temporal logics—is satisfied.

Theorem 3 is critical as it provides soundness guarantees, ensuring that the policy derived from the skill machine will always satisfice the task requirements. Finally, in cases where the composed skill $\delta_Q$ obtained from the approximate SM is not sufficiently optimal, we can use any off-policy algorithm to learn a new skill $Q_\mathcal{T}$ few-shot. This is achieved by using the maximising Q-values $\max\{\beta Q_\mathcal{T}, (1-\beta)\delta_Q\}$ in the exploration policy during learning. Here, $\beta \in (0, 1)$ is a parameter that determines how much of the composed policy to use. It can also be seen as decreasing the potentially overestimated values of $\delta_Q$, since $\delta_Q$ is greedy with respect to both goals and RM transitions. Consider Q-learning with $\beta = \gamma$. During the $\epsilon$-greedy exploration, we use $a \leftarrow \arg\max_\mathcal{A} \max\{\gamma Q_\mathcal{T}, (1-\gamma)\delta_Q\}$ to select greedy actions, hence improving the initial performance of the agent where $\gamma Q_\mathcal{T} < (1 - \gamma)\delta_Q$, and guaranteeing convergence in the limit like regular Q-learning. Appendix A.2 illustrates this process.

## 4 Experiments

We consider the Office Gridworld (Figure 1a) and the Moving Targets (Figure 4) domains: (i) **Office Gridworld Icarte et al. (2018):** The tasks here are specified over 10 propositions $\mathcal{P} = \{A, B, C, D, \maltese, \text{☕}, \boxtimes, \text{♟}, \boxtimes^+, \text{♟}^+\}$ and 1 constraint $\mathcal{C} = \{\maltese\}$. We learn the base skill primitives $\mathcal{Q}_\mathcal{P}^*$ (Figure 7 in Appendix A.5) using goal oriented Q-learning Nangue Tasse et al. (2020), where the agent keeps track of reached goals and uses Q-learning (Watkins, 1989) to update the WVF with respect to all seen goals at every time step. (ii) **Moving Targets Domain Nangue Tasse et al. (2020):** This is a canonical object collection domain with high dimensional pixel observations ($84 \times 84 \times 3$ RGB images). The agent here needs to pick up objects of various shapes and colors; picked objects respawn at random empty positions similarly to previous object collection domains (Barreto et al., 2020). There are 3 object colours—beige (□), blue (■), purple (■)—and 2 object shapes—squares (⊠), circles (○). The tasks here are defined over 6 propositions and constraints $\mathcal{P} = \mathcal{C} = \{□, ■, ■, ⊠, ○\}$. We learn the corresponding base skill primitives with goal oriented Q-learning Nangue Tasse et al. (2020) but using deep Q-learning (Mnih et al., 2015) to update the WVFs.

### 4.1 Zero-shot temporal logics

| Task | Description \| LTL |
|------|--------------------|
| 1 | Deliver coffee to the office without breaking decorations $\mid \left(F\left(\text{☕} \wedge X\left(F\,\text{♟}\right)\right)\right) \wedge (G\,\neg\maltese)$ |
| 2 | Patrol rooms A, B, C, and D without breaking any decoration $\mid \left(F\left(\mathbf{A} \wedge X\left(F\left(\mathbf{B} \wedge X\left(F\left(\mathbf{C} \wedge X\left(F\mathbf{D}\right)\right)\right)\right)\right)\right)\right) \wedge (G\,\neg\maltese)$ |
| 3 | Deliver coffee and mail to the office without breaking any decoration $\mid \left(\left(F\left(\text{☕} \wedge X\left(F\left(\boxtimes \wedge X\left(F\text{♟}\right)\right)\right)\right)\right) \| \left(F\left(\boxtimes \wedge X\left(F\left(\wedge X\left(F\text{♟}\right)\right)\right)\right)\right)\right) \wedge (G\neg\maltese)$ |
| 4 | Deliver mail to the office until there is no mail left, then deliver coffee to office while there are people in the office, then patrol rooms A-B-C-D-A, and never break a decoration $\mid \left(F\left(\boxtimes \wedge X\left(F\left(\text{♟} \wedge X\left(\neg\boxtimes U\left(\neg\boxtimes^+ \wedge \boxtimes \wedge X\left(F\left(\text{☕} \wedge X\left(\neg\text{♟}U\left(\neg\text{♟}^+ \wedge \text{♟} \wedge X\right.\right.\right.\right.\right.\right.\right.\right.\right.\right.$ $\left.\left.\left.\left.\left.\left.\left.\left.\left.\left.\left.(FA \wedge X\left(F\left(B \wedge X\left(F\left(C \wedge X\left(F\left(D \wedge X\left(FA\right)\right)\right)\right)\right)\right)\right)\right)\right)\right)\right)\right)\right)\right)\right)\right)\right)\right) \wedge (G\,\neg\maltese)$ |

Table 1: Tasks in the Office Gridworld. The RMs are generated from the LTL expressions.

We use the Office Gridworld as a multitask domain, and we evaluate how long it takes an agent to learn a policy that can solve the four tasks described in Table 3. The agent iterates through the tasks, changing from one to the next after each episode. In all of our experiments, we compare the performance of skill machines with that of state-of-the-art RM-based learning approaches like counterfactual RMs (CRM)—where the Q-functions are updated with respect to all possible RM transitions from a given environment state—and hierarchical RMs (HRM)—where an agent learns options per RM state that are grounded in the environment states (Icarte et al., 2018). In addition to learning all four tasks, we also experiment with Tasks 3 and 4 in isolation. In these single task domains, the difference between CRM, HRM, skill machines and Q-learning should be less pronounced, since CRM, HRM and few-shot with skill machines now cannot leverage the shared experience across multiple tasks. Thus, the comparison between multi-task and single-task learning in this setting will evaluate the benefit of the compositionality afforded by skill machines.

The results of these three experiments (each ran for $2 \times 10^5$ time steps) are shown in Figure 3. Regular Q-learning struggles to learn Task 3 and completely fails to learn the hardest task (Task 4).

Additionally, notice that while QL and CRM can theoretically learn the tasks optimally given infinite time, only HRM and SM are able to learn hard long horizon tasks in practice. It is important to note that we train all algorithms for the same amount of time during these experiments and previous work (Nangue Tasse et al., 2020) has shown that learning the WVFs takes longer than learning task-specific skills. In addition, the skill machines are being used to zero-shot generalise to the office tasks using skill primitives. Thus using the skill machines in isolation (labelled SM on Figure 3) may provide sub-optimal performance compared to the task-specific agents, since the skill machines have not been trained to optimality and are not specialised to the domain. Even under these conditions, we observe that skill machines perform near-optimally in terms of final performance, and due to the amortised nature of learning the WVF will achieve its final rewards from the first epoch.

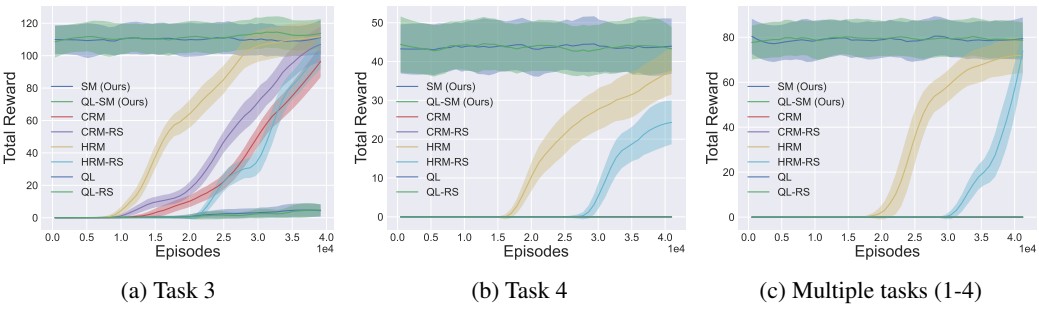

(a) Task 3                  (b) Task 4               (c) Multiple tasks (1-4)

Figure 3: Average (over 80 independent trials) returns during training in the Office Gridworld.

## 4.2 Few-shot Temporal Logics

It is possible to pair the skill machines with a learning algorithm such as Q-learning to achieve few-shot generalisation. From the results shown in Figure 3, it is apparent that skill machines paired with Q-learning (labelled QL-SM on Figure 3) achieves the best performance for both the single-task and multi-task setting. While it is not clear from the rewards that adding Q-learning provides significant improvements to the skill machine, their trajectories show that Q-learning does indeed improve on the skill machine policies when they are not optimal (Appendix 9). Additionally, skill machines with Q-learning always begin with a significantly higher reward and converge on their final performance faster than all benchmarks—except the zero-shot one which is (near) optimal in all cases. The speed of learning is due to the compositionality of the skill primitives with skill machines, and the high final performance is due to the generality of the learned primitives being paired with the domain specific Q-learner. In sum, skill machines provide fast composition of skills and achieve optimal performance compared to all benchmarks when paired with a learning algorithm.

## 4.3 Function Approximation

We now demonstrate our temporal logic composition approach in Moving Targets domain where function approximation is required. Figure 4 shows the average returns of the optimal policies and SM policies for the four tasks described in Table 2 with a maximum of 50 steps per episode. Our results show that even when using function approximation with sub-optimal skill primitives, the zero-shot policies obtained from skill machines are very close to optimal on average. We also observe that for very challenging tasks like Tasks 3 and 4 (where the agent must satisfice difficult temporal constraints), the

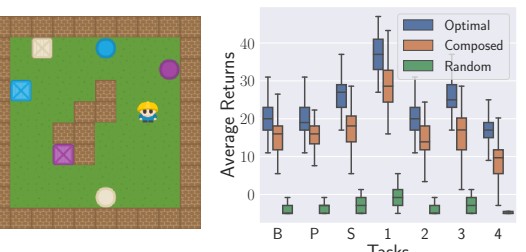

Figure 4: The Moving Targets domain (left) and the average returns over 100 runs for tasks in Table 2 (right), where $B, P, S = $ ■, ■, ⊠.

compounding effect of the sub-optimal policies sometimes leads to failures. In such cases, learning new skills few-shot using tabular Q-learning by leveraging the SM would guarantee convergence to optimal policies as demonstrated in Section 4.2, but that is not guaranteed using function approximation.

| Task | Description \| LTL |
|------|-------------------|
| 1 | Pick up any object. Repeat this forever. \| $F(\bigcirc \vee \boxtimes)$ |
| 2 | Pick up blue then purple objects, then objects that are neither blue nor purple. Repeat this forever. \| $F(\blacksquare \wedge X(F(\blacksquare \wedge X(F((\bigcirc \vee \boxtimes) \wedge \neg(\blacksquare \vee \blacksquare))))))$ |
| 3 | Pick up blue objects or squares, but never blue squares. Repeat this forever. \| $(F(\blacksquare \vee \boxtimes)) \wedge (G \neg(\blacksquare \wedge \boxtimes))$ |
| 4 | Pick up non-square blue objects, then non-blue squares in that order. Repeat this forever. \| $F((\neg\boxtimes \wedge \blacksquare) \wedge X(F(\boxtimes \wedge \neg\blacksquare)))$ |

Table 2: Tasks in the Moving Targets domain. To repeat forever, the terminal states of the RMs generated from LTL are removed, and transitions to them are looped back to the start state.

## 5 RELATED WORK

One family of approaches to spatial composition leverages forms of regularisation to achieve semantically meaningful disjunction (Todorov, 2009; Van Niekerk et al., 2019) or conjunction (Haarnoja et al., 2018; Hunt et al., 2019). Weighted composition has also been demonstrated; for example, Peng et al. (2019) learn weights to compose existing policies multiplicatively to solve new tasks. Approaches that leverage the successor feature (SF) framework (Barreto et al., 2017) are capable of solving tasks defined by linear preferences over features (Barreto et al., 2020). Alver & Precup (2022) show that an SF basis can be learned that is sufficient to span the space of tasks under consideration, while Nemecek & Parr (2021) determine which policies should be stored in limited memory so as to maximise performance on future tasks. In contrast to these approaches, our framework allows for both spatial composition (including operators such as negation that other approaches do not support) and temporal composition such as LTL.

A popular way of achieving temporal composition is through the options framework (Sutton et al., 1999; Bacon et al., 2017). Here, high-level skills are first discovered and then executed sequentially to solve a task (Konidaris & Barto, 2009; Bagaria & Konidaris, 2019). Barreto et al. (2019) leverage the SF and options framework and learn how to linearly combine skills, chaining them sequentially to solve temporal tasks. However, these options-based approaches offer a relatively simple form of temporal composition. By contrast, we are able to solve tasks expressed through regular languages zero-shot, while providing soundness guarantees.

Work has also centred on approaches to defining tasks using human-readable logic operators. For example, Li et al. (2017) and Littman et al. (2017) specify tasks using LTL, which is then used to generate a standard reward signal for an RL agent. Camacho et al. (2019) show how to perform reward shaping given LTL specifications, while Jothimurugan et al. (2019) develop a formal language that encodes tasks as sequences, conjunctions and disjunctions of subtasks. This is then used to obtain a shaped reward function that can be used for learning. All of these approaches focus on how an agent can improve learning given such specifications or structure, but we show how an explicitly compositional agent can immediately solve such tasks using WVFs without further learning.

## 6 CONCLUSION

We proposed skill machines—finite state machines that can be learned from reward machines—that allow agents to solve extremely complex tasks involving temporal and spatial composition. We demonstrated how skills can be learned and encoded in a specific form of goal-oriented value function that, when combined with the learned skill machines, are sufficient for solving subsequent tasks without further learning. Our approach guarantees that the resulting policy adheres to the logical task specification, which provides assurances of safety and verifiability to the agent's decision making, important characteristics that are necessary if we are to ever deploy RL agents in the real world. While the resulting behaviour is provably satisficing, empirical results demonstrate that the agent's performance is near optimal; further fine-tuning can be performed should optimality be required, which greatly improves the sample efficiency. We see this approach as a step towards truly generally intelligent agents, capable of immediately solving human-specifiable tasks in the real world with no further learning.

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

## A  APPENDIX

### A.1  PROOFS OF THEORETICAL RESULTS

**Theorem 1.** *Let $\mathbf{R}_{\mathcal{G}}$ be a vector of rewards for each task primitive, and $\bar{\mathbf{Q}}_{\mathcal{G}}^*$ be the corresponding vector of optimal WVFs. Then, for an MDP $m = \langle \mathcal{S}_{\mathcal{G}}, \mathcal{A}_{\mathcal{G}}, P_{\mathcal{G}}, R_{\mathbf{w}}, \gamma \rangle$ with linear preference reward function $R_{\mathbf{w}} = \mathbf{w} \cdot \mathbf{R}_{\mathcal{G}}$, we have*

$$\bar{Q}_m^* = \mathbf{w} \cdot \bar{\mathbf{Q}}_{\mathcal{G}}^*.$$

*Proof.*

$$\bar{Q}_m^*(s, g, a) = \mathbb{E}_{\bar{\pi}^*}\left[\sum_{t=0}^{\infty} \gamma^t \mathbf{w} \cdot \bar{\mathbf{R}}_{\mathcal{G}}(s_t, g, a_t)\right]$$

$$= \mathbf{w} \cdot \mathbb{E}_{\bar{\pi}^*}\left[\sum_{t=0}^{\infty} \gamma^t \bar{\mathbf{R}}_{\mathcal{G}}(s_t, g, a_t)\right];$$

since the world policies are independent of task Nangue Tasse et al. (2020)[Lemma 2].

$$= \mathbf{w} \cdot \bar{\mathbf{Q}}_{\mathcal{G}}^*$$

$\square$

**Theorem 2.** *Let $\pi^*(s, u)$ be the optimal policy for a task $M_{\mathcal{T}}$, and let $\mathcal{C} = \mathcal{P}$. Then there exists a corresponding skill machine with a $\mathbf{w}_{\mathcal{G}}$ and $\mathbf{w}_U$ such that*

$$\pi^*(s, u) \in \underset{a \in \mathcal{A}}{\arg\max} \, \delta_Q(u)(\langle s, c \rangle, \langle a, 0 \rangle),$$

*where $\delta_Q$ is given by $\mathbf{w}_{\mathcal{G}}$ and $\mathbf{w}_U$ as per Definition 5.*

*Proof.* Let $\mathbf{w}_U(u, \cdot) = \frac{1}{N_{\delta_u}}$ where $N_{\delta_u}$ is the number of possible RM transitions from $u$. Also let $\mathbf{w}_{\mathcal{G}}(s, u, \cdot)$ be 1 for the set of propositions $g \in 2^{\mathcal{C}}$ that are satisfied when following $\pi^*(s, u)$, and zero everywhere else. Then $\pi^*(s, u) \in \arg\max_{a \in \mathcal{A}} \delta_Q(u)(\langle s, c \rangle, \langle a, 0 \rangle)$ since $\mathbf{w}_U(u, u') \bar{Q}_{u,u'}^*(\langle s, c \rangle, g, \langle a, 0 \rangle)$ is optimal using Theorem 1 and optimal policies are assumed to reach task goals. $\square$

**Theorem 3.** *Let $R_{\mathcal{PA}} = \langle U, u_0, F, \delta_u, \delta_{\mathcal{M}} \rangle$ be a satisfice-able simple reward machine with non-zero rewards $R_{MAX}$ only for accepting transitions, and for which all valid transitions $(u, u')$ are achievable from any state $s \in \mathcal{S}$. Define the skill machine $\bar{\mathcal{Q}}_{\mathcal{SA}}^* = \langle U, u_0, F, \delta_u, \delta_Q, \mathbf{w}_U, \mathbf{w}_{\mathcal{G}} \rangle$ with*

$$\mathbf{w}_U(u, u') := 1 \quad \text{if } u' = \underset{u''}{\arg\max} \, Q^*(u, u''), \quad 0 \quad \text{otherwise}$$

$$\mathbf{w}_{\mathcal{G}}(\langle s, c \rangle, u, g) := 1 \quad \text{if } g = \underset{g'}{\arg\max} \max_a \sum_{u'} \mathbf{w}_U(u, u') \bar{Q}_{u,u'}^*(\langle s, c \rangle, g', \langle a, 0 \rangle), \quad 0 \quad \text{otherwise}$$

*where $Q^*$ is the optimal transition-value function for $R_{\mathcal{PA}}$. Then following the policy $\pi(s, u) \in \arg\max_{a \in \mathcal{A}} \delta_Q(u)(\langle s, c \rangle, \langle a, 0 \rangle)$, will reach an accepting transition.*

*Proof.* This follows from the optimality of $\pi^*(s, u)$ and $Q^*$, since each transition of the RM is satisfice-able from any environment state. $\square$

## A.2 Pseudo-Code for Few-shot Q-learning using Skill Machines

---

**Algorithm 1:** Few-shot Q-learning using skill machines

---

**Input** : $\gamma, \alpha, \mathcal{P}, \mathcal{C}, L, U, u_0, F, \delta_u, \delta_Q$
**Initialise :** $Q(s, u, a)$
**foreach** *episode* **do**
    Observe initial state $s \in \mathcal{S}$, get initial $u \leftarrow u_0$, and $c = 0$
    **while** *episode is not done* **do**
        /* Using the composed skill $\delta_Q$ in the behaviour policy   */
        $a \leftarrow$
$$\begin{cases} \underset{a \in \mathcal{A}}{\arg\max} \left( \max\{\gamma Q(s, u, a), (1 - \gamma)\delta_Q(u)(\langle s, c\rangle, \langle a, 0\rangle)\} \right) \text{ if } Bernoulli(1 - \epsilon) = 1 \\ \text{a random action} \quad \text{otherwise} \end{cases}$$

        Take action $a$ and observe next state $s'$ and true constraints $c \leftarrow c \cup (\mathcal{C} \cap L(s, a, s'))$
        Get reward $r \leftarrow \delta_r(u)(s, a, s')$ and the next RM state $u' \leftarrow \delta_u(u, L(s, a, s'))$
        $Q(s, u, a) \xleftarrow{\alpha} r$ **if** $s'$ is terminal or $u' \in F$ **else** $\left[ r + \gamma \underset{a'}{\max} Q(s', u', a') \right]$
        $s \leftarrow s'$

---

## A.3 Function Approximation with Continuous Actions and States

| Task | Description \| LTL |
|------|-----------------|
| 1 | Navigate to a button and then to a cylinder. \| $(F\,(\mathbf{B} \wedge X\,(F\,\mathbf{C})))$ |
| 2 | Navigate to a button and then to a cylinder while never entering hazard regions \| $(F\,(\mathbf{B} \wedge X\,(F\,\mathbf{C}))) \wedge (G\,\neg\mathbf{H})$ |
| 3 | Navigate to a button, then to a cylinder without entering hazard regions, then to a button inside a hazard region, and finally to a cylinder again. \| $F\,(\mathbf{B} \wedge X\,(F\,((\mathbf{C} \wedge \neg\mathbf{H}) \wedge X\,(F\,((\mathbf{B} \wedge \mathbf{H}) \wedge X(F\mathbf{H}))))))$ |
| 4 | Navigate to a button and then to a cylinder in a hazard region. \| $(F\,(\mathbf{B} \wedge X\,(F\,\mathbf{C} \wedge \mathbf{H})))$ |
| 5 | Navigate to a cylinder, then to a button in a hazard region, and finally to a cylinder again. \| $(F\,(\mathbf{C} \wedge X\,(F\,((\mathbf{B} \wedge \mathbf{H}) \wedge X\,(\mathbf{C}))))$ |
| 6 | Navigate to a hazard, then to a cylinder, and finally to a cylinder again while avoiding hazards. \| $(F\,(\mathbf{H} \wedge X\,(F\,(\mathbf{C} \wedge X\,(F\,(\mathbf{C} \wedge \mathbf{H}))))))$ |

Table 3: Tasks in the Safety AI Gym domains. The RMs are generated from the LTL expressions.

We demonstrate our temporal logic composition approach in a Safety AI Gym domain (Figure 5) (Ray et al., 2019) which has a continuous state space ($\mathcal{S} = \mathbb{R}^{60}$) and continuous action space ($\mathcal{A} = \mathbb{R}^2$). The agent here is a point mass that needs to navigate to various regions defined by 3 propositions ($\mathcal{P} = \{B, C, H\}$) corresponding to its 3 lidar sensors for the *buttons* ($B$) (grey spheres), the *cylinder* ($C$) (translucent cylinder), and the *hazards* ($H$) (blue regions). The button and hazard positions are fixed as shown in Figure 5, the cylinder is randomly placed on one of the buttons, and the agent is randomly placed anywhere on the plane. We first learn the 3 base skill primitives corresponding to each predicate (with constraints $\mathcal{C} = \{H\}$), with goal oriented Q-learning Nangue Tasse et al. (2020) but using Twin Delayed DDPG (Fujimoto et al., 2018) to update the WVFs. Figure 6 shows the trajectories of the SM policies for the six tasks described in Table 3. Our results shows that skill primitives can be leveraged to achieve zero-shot temporal logics even in continuous domains.

## A.4 Details of Experimental Setting

In this section we elaborate further on the hyper-parameters for the various experiments in Section 4. We also describe the pretraining of WVFs for all of the experimental settings which corresponds to learning the base task primitives for each domain. The same hyper-parameters are used for all algorithms in a particular experiment. This is to ensure that we evaluate the relative performance

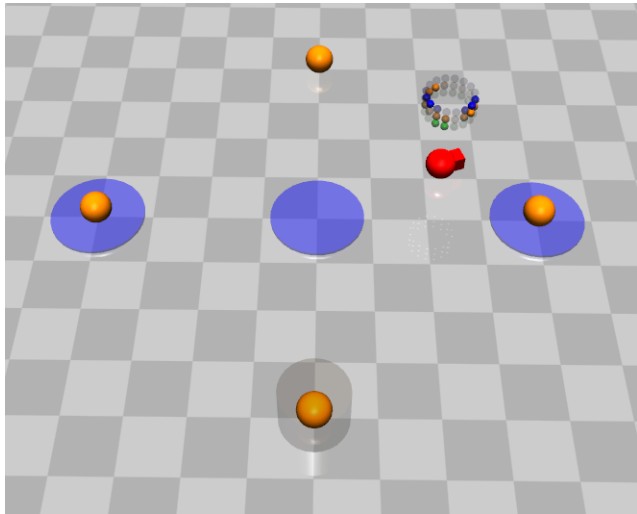

Figure 5: Visualisation of the Safety AI Gym Domain.

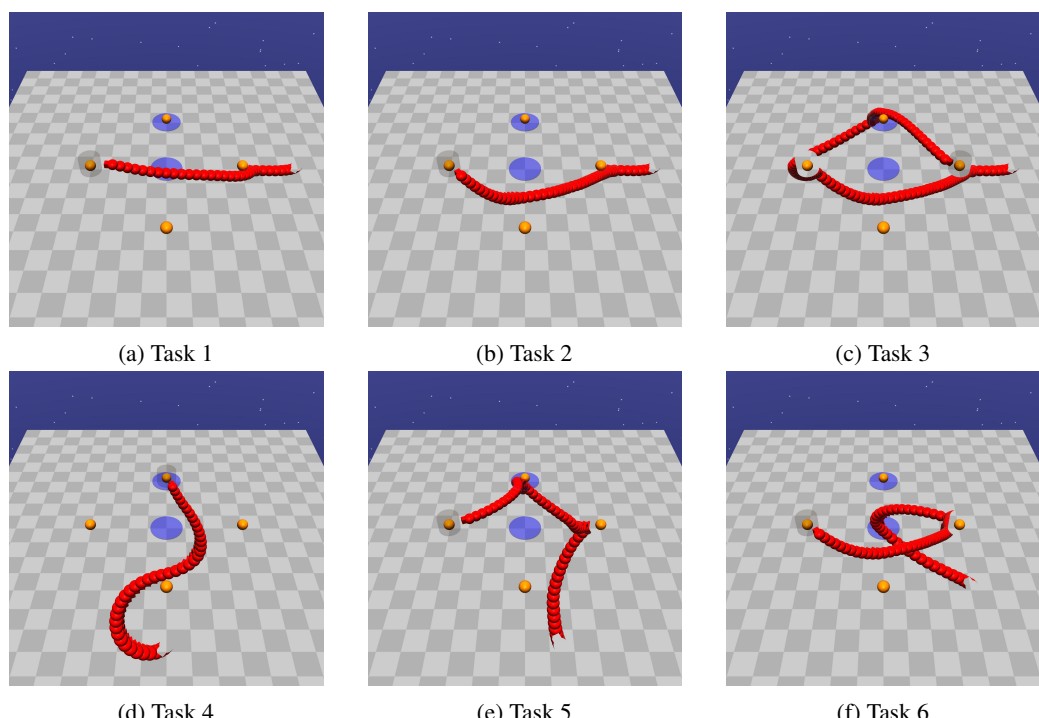

| (a) Task 1 | (b) Task 2 | (c) Task 3 |
| (d) Task 4 | (e) Task 5 | (f) Task 6 |

Figure 6: Visualisations of the trajectories obtained by following the zero-shot composed policies from the skill machine for tasks in Table 3.

fairly and consistently, particularly since Nangue Tasse et al. (2020) note that learning WVFs can take longer than direct training of tasks. However, training WVFs for longer would potentially bias the results in favour of their use. Thus, all algorithms are trained for the same amount of time which is set such that they all converge. The full list of hyper-parameters for the Office World, Moving Targets and SafeAI Gym domain experiments are shown in Tables respectively.

To use skill machines we require pre-trained WVFs. As mentioned above, all WVFs are trained using the same hyper-parameters as any other training. Additionally, for all experiments the WVFs are pre-trained on the base task primitives for the domain. For example, in the Office World domain, the WVFs are trained on the $\mathcal{P}$ base task primitive corresponding to achieving each predicate, $\mathcal{P} =$

| Hyper-parameter | Value |
|---|---|
| Timesteps | $2e^5$ |
| Training exploration ($\epsilon$) | 0.5 |
| Per-episode evaluation exploration ($\epsilon$) | 0.1 |
| Discount Factor ($\gamma$) | 0.9 |

Table 4: Table of hyper-parameters used for Q-learning in the Office World experiments.

| Hyper-parameter | Value |
|---|---|
| Timesteps | $1e^6$ |
| Neural Network architecture | $CNN + MLP$ |
| CNN architecture | Defaults of Mnih et al. (2015) |
| MLP hidden layers | $1024 \times 1024 \times 1024$ |
| Start exploration ($\epsilon$) | 1 |
| End exploration ($\epsilon$) | 0.1 |
| Exploration decay duration ($\epsilon$) | $5e^5$ |
| Discount Factor ($\gamma$) | 0.99 |
| Others | Defaults of Mnih et al. (2015) |

Table 5: Table of hyper-parameters used for Deep Q-learning in the Moving Targets experiments.

| Hyper-parameter | Value |
|---|---|
| Timesteps | $1e^6$ |
| Neural Network architecture | $MLP$ |
| MLP hidden layers | $2024 \times 2024 \times 2024$ |
| Max episodes length | 100 |
| Target noise | 0.2 |
| Action noise | 0.2 |
| Discount Factor ($\gamma$) | 0.99 |
| Others | Defaults of Achiam (2018) |

Table 6: Table of hyper-parameters used for the TD3 in the SafeAI Gym experiments.

$\{A, B, C, D, \maltese, \textrm{☕}, \boxtimes, \textrm{♟}, \boxtimes^+, \textrm{♟}^+\}$ (reaching states the predicate is set to True), with constraints $\mathcal{C} = \{\maltese\}$. All other primitives in this domain can be obtained zero-shot through value function composition. Similarly, for the moving targets domain, the WVFs are pre-trained on the primitives corresponding to obtaining objects by shape or colour in the environment separately, $\mathcal{P} = \{\square, \blacksquare, \blacksquare, \boxtimes, \bigcirc\}$, with constraints $\mathcal{C} = \mathcal{P}$. From here the value functions for finding objects of particular colours or any more complex primitives can be composed zero-shot. Finally, for the SafeAI Gym environment the base skill primitives correspond to going to a $button$ $(B)$, a $cylinder$ $(C)$, and a $hazard$ $(H)$: $\mathcal{P} = \{B, C, H\}$, trained with constraints $\mathcal{C} = \{H\}$.

A.5    OFFICE WORLD ADDITIONAL FIGURES

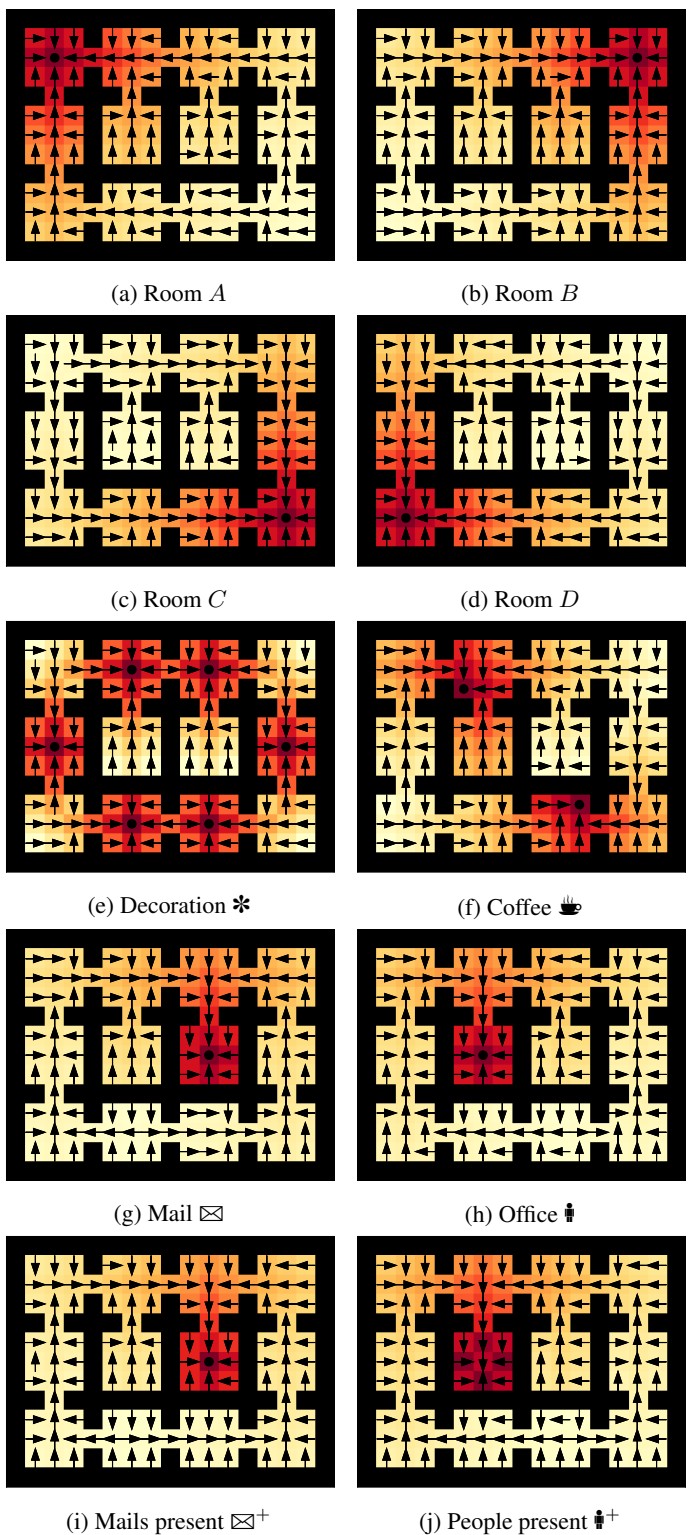

(a) Room $A$

(b) Room $B$

(c) Room $C$

(d) Room $D$

(e) Decoration ✲

(f) Coffee ☕

(g) Mail ✉

(h) Office 👤

(i) Mails present ✉$^+$

(j) People present 👤$^+$

Figure 7: The policies (arrows) and value functions (heat map) of the base primitive tasks in the Office Gridworld. These are obtained by maximising over the goals of the learned WVFs. All errors in the figures are due to training the WVFs for 200000 time steps, hence not to convergence.

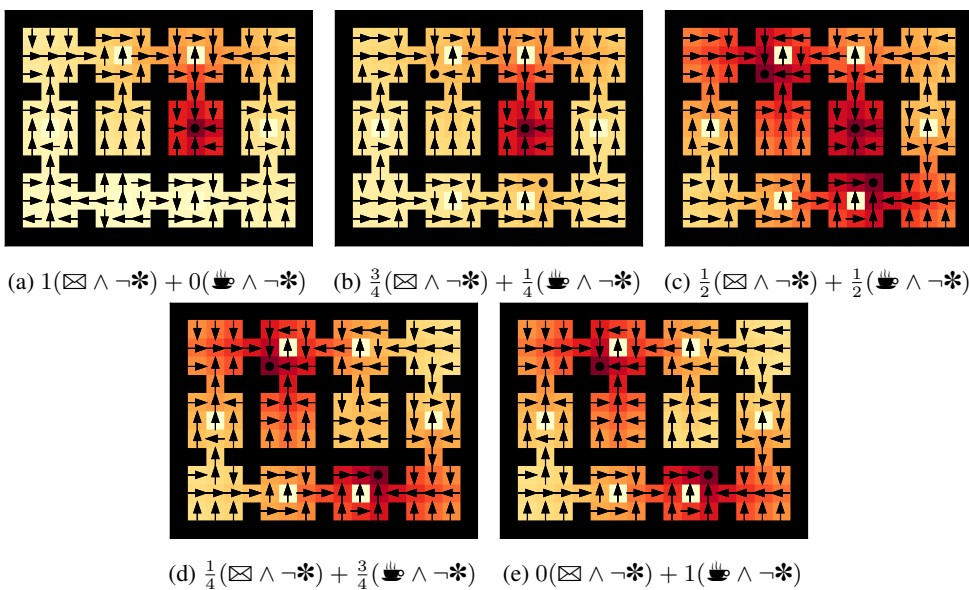

(a) $1(\boxtimes \wedge \neg \maltese) + 0(\text{☕} \wedge \neg \maltese)$   (b) $\frac{3}{4}(\boxtimes \wedge \neg \maltese) + \frac{1}{4}(\text{☕} \wedge \neg \maltese)$   (c) $\frac{1}{2}(\boxtimes \wedge \neg \maltese) + \frac{1}{2}(\text{☕} \wedge \neg \maltese)$

(d) $\frac{1}{4}(\boxtimes \wedge \neg \maltese) + \frac{3}{4}(\text{☕} \wedge \neg \maltese)$   (e) $0(\boxtimes \wedge \neg \maltese) + 1(\text{☕} \wedge \neg \maltese)$

Figure 8: The policies (arrows) and value functions (heat map) for various preferences over the primitives "get a mail without breaking decorations" ($\boxtimes \wedge \neg \maltese$) and "get coffee without breaking decorations" ($\text{☕} \wedge \neg \maltese$) in the Office GridWorld. These are obtained by first doing the weighted sum of the composed WVFs ($\bar{Q}^*_{\boxtimes} \wedge \neg \bar{Q}^*_{\maltese}, \bar{Q}^*_{\text{☕}} \wedge \neg \bar{Q}^*_{\maltese}$) according to the preferences, and then maximising over the goals and actions. All errors in the figures are due to training the WVFs of the base primitives ($\bar{Q}^*_{\text{☕}}, \bar{Q}^*_{\boxtimes}$) for 200000 time steps, hence not to convergence.

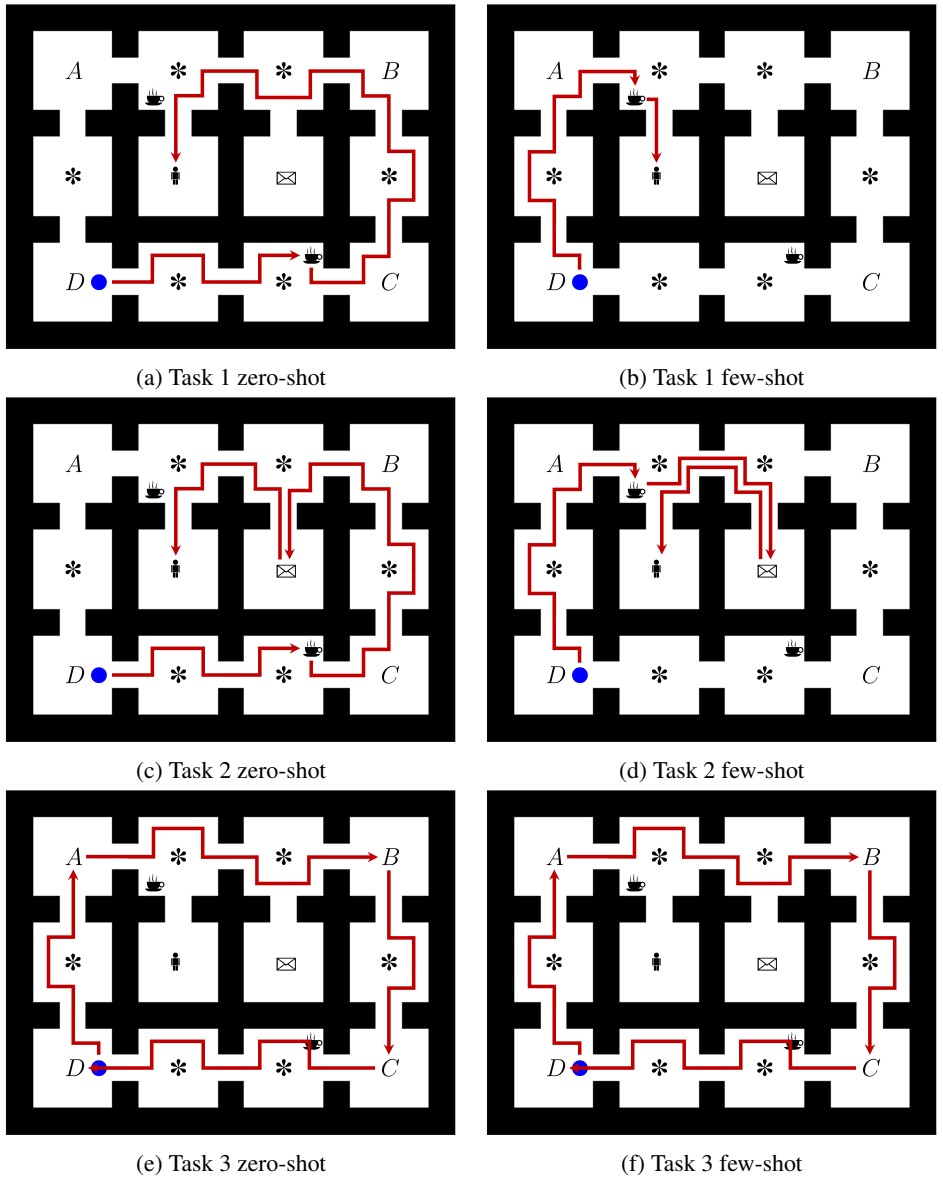

Figure 9: Agent trajectories for various tasks in the Office Gridworld (Table 3) using the skill machine without further learning (left) and with further learning (right).

