# OpenReview forum: "Skill Machines: Temporal Logic Composition in Reinforcement Learning"
_ICLR.cc/2023/Conference — Submitted to ICLR 2023_

### Official Review · Reviewer_MuPq · 2022-10-24

**Confidence:** 4
**Correctness:** 3
**Technical Novelty And Significance:** 3
**Empirical Novelty And Significance:** 2
**Recommendation:** 6

**Clarity, Quality, Novelty And Reproducibility:**

There are some technically inaccurate statements in the paper. Formally, "regular language" is a particular class of languages and saying that "any regular language, such as linear temporal logics" is inaccurate. LTL is not in RE.  Also, "video game environment" can be rewritten as "game environment".

At some point in the paper, it states that "We therefore only need concern ourselves with how to solve any task expressed as a linear combination of the primitive tasks" . It is not clear why any task would correspond to such a linear combination.





**Strength And Weaknesses:**

Strengths:

- The idea of learning individual skills separately from the task-specific policy is very interesting.

- The combination of individual skills with reward machines can be very effective.

Weaknesses:

- There is a major assumption in the paper that needs to be more clearly stated. The considered multitask setting assumes that the underlying MDP can be designed such that the reward for any specific task is R0 + R_\tau where R0 is default reward and R_\tau is reward for a specific task tau. This is not necessarily true, and some work on IRL https://proceedings.neurips.cc/paper/2018/hash/74934548253bcab8490ebd74afed7031-Abstract.html have noticed this to recognize the value of learning temporal specifications as rewards (because this simple sum decomposition is not always feasible). This does not weaken the main contribution of the paper but it is important to recognize the significance of this assumption.

- The tasks's state space is defined to be product of the environment state and the set of constraints incorporating the set of propositions that are currently true. The latter set can be exponential, and this seems to indicate an exponential blow-up in the state space with the growth in the size of the specifications.

- The experiments seem to use mostly planning tasks. Using examples from Safe AI Gym would be useful to evaluate the full potenital of this method since it is being represented as a general RL approach and not just something limited to discrete planning.

**Summary Of The Paper:**

The paper proposes a new RL framework wherein a set of base skills are learned in a reward-free setting followed by combination of these skills and learning skill machine to produce a complex behavior specified using languages such as linear temporal logic. This allows zero-shot mapping from high-level LTL specification to complex behavior. This is particularly helpful in solving long horizon task n the presence of a sparse learning signal.

**Summary Of The Review:**

Overall, the approach presented in the paper is interesting, but its explanation has some gaps, making the reviewer unable to appreciate it fully. Clarification to the doubts expressed in the review would be useful.

The reviewer has requested some minor revisions for technical accuracy and would encourage a careful review of some statements. Overall, the paper has enough merit to deserve a positive score and the reviewer has raised the rating.

---

> ### Author Response · Authors · 2022-11-17
> **Thanks for the thorough review. We have updated the paper to clarify your concerns.**
>
> Thanks for your thorough review of our paper. Also thanks to your concerns and that of the other reviewers, we have updated the paper to clarify our notations and the definitions of tasks (and how their rewards per RM nodes are defined as linear preferences over Boolean expressions, Def 3), and primitives (Def 4). We believe the paper is much clearer now and hope it addresses most of your concerns. We hope to have also addressed your specific points bellow:
>
> > About the assumption that the reward for any specific task is $R_0 + R_\tau$
>    * The $R_0 + R_\tau$ assumption is only for the rewards per state of the reward machine. In the simplest case, $R_0 = 0$ and  $R_\tau = 1$ at goal states and 0 everywhere else, as in the suggested paper [1].
>    * The tasks specified by finite state machines (reward machines in our case) are indeed not necessarily Markovian, as seen by the examples in Figure 1 or 2, and also as seen in the paper you suggested which learns such specifications from demonstrations.
>    * Hence, our framework works in the more general case where the rewards per transition from the reward machine specifications are of the form $R_0 + R_\tau$, instead of just binary (e.g 0 or 1) as in the suggested paper.
>
> > The set of constrains indicate an exponential blow-up in the state space with the growth in the size of the specifications.
>    * Indeed, this is a fundamental tradeoff in our approach, since the skill primitives learned with those constraints enable agents to solve temporal logic tasks zero-shot. Otherwise, one would have to learn most new tasks, of which there are an infinite number and their reward machines can be arbitrarily complex (making training time extremely long and hence impractical as demonstrated by our experiments).
>
> > The experiments seem to use mostly planning tasks. Using examples from Safe AI Gym would be useful to evaluate the full potenital of this method since it is being represented as a general RL approach and not just something limited to discrete planning.
>    * We note that this work is focused on tasks specified as reward machines.
>    * Hence for any domain, we can always learn the skill primitives by using any suitable RL algorithm (in the primitives MDPs, Def 2), and plan over the simple RMs to obtain skill-machines.
>    * We use similar domains to previous works [2,3,4] since they are representative of environments with combinatorially large number of tasks and still sufficiently hard. Similarly to those works, our approach is agnostic to the specific environment (i.e any suitable RL algorithm can be used to learn the value functions in the defined MDPs).
>    * For example, our function approximation experiments (the moving targets domain) have 84x84x3 RGB pixel observations, and we simply use a DQN to learn the skill primitives. Similarly, for a given set of propositions/sensors in a Safe AI Gym environment (e.g goal, hazards, buttons, ...), the corresponding skill primitives can be learned with an appropriate RL algorithm.
>    * Thanks for the Safe AI Gym suggestion. We are running the same function approximation experiments in it and will update the paper with the results.
>
> > "any regular language, such as linear temporal logics" is inaccurate. LTL is not in RE
>    * Thanks for noticing that. It is technically an omega-regular language.
>    * We meant here that both REs and LTLs can be converted to finite state automata (for example Buchi automata for LTL).
>    * For simplicity, we have rephrased that in the abstract to "any regular language and even linear temporal logics".
>
> >  The "video game environment" can be rewritten as "game environment"
>    * “Video” is important here because we are using high-dimensional (84x84x3) RGB pixel observations.
>
> > It is not clear why any task would correspond to such a linear combination.
>    * That is for the rewards at each state of the reward machine. In the simplest case the linear preference could be 1 for a given Boolean expression (e.g coffee and not decorations) and 0 for all others, which will mean that the rewards from the reward machine are binary (0 or 1) as in the paper you suggested earlier [1].
>
>
> [1]  https://proceedings.neurips.cc/paper/2018/hash/74934548253bcab8490ebd74afed7031-Abstract.html
>
> [2] Geraud Nangue Tasse, Steven James, and Benjamin Rosman. A Boolean task algebra for reinforcement learning.
>
> [3] Rodrigo Toro Icarte, Toryn Klassen, Richard Valenzano, and Sheila McIlraith. Using reward machines for high-level task specification and decomposition in reinforcement learning.
>
> [4] André Barreto, Shaobo Hou, Diana Borsa, David Silver, and Doina Precup. Fast reinforcement learning with generalized policy updates

---

> > ### Comment · Reviewer_MuPq · 2022-11-30
> > **Thank you; score has been raised**
> >
> > > Thanks for the Safe AI Gym suggestion. We are running the same function approximation experiments in it and will update the paper with the results.
> >
> > The reviewer thanks the authors for accepting this request and is raising the score.
> >
> > > That is for the rewards at each state of the reward machine. In the simplest case the linear preference could be 1 for a given Boolean expression (e.g coffee and not decorations) and 0 for all others, which will mean that the rewards from the reward machine are binary (0 or 1).
> >
> > This simplest case would allow sequential combination of tasks where only one task is active, but the description in the paper appeared to hint to a more general composition. The reviewer does not suggest that linear combination is not useful, but rather that the limitation of what kinds of compositions can be captured (and what cannot be captured) by linear combination needs to be discussed. The concern is that the linear combination is going to be very restrictive.
> >
> > > "any regular language, such as linear temporal logics" is inaccurate. LTL is not in RE
> > > Thanks for noticing that. It is technically an omega-regular language.
> > > We meant here that both REs and LTLs can be converted to finite state automata (for example Buchi automata for LTL).
> >
> > No, LTL cannot be represented by a finite state automata similar to REs. There is some serious misunderstanding here. Buchi automata accept/reject infinite strings. The paper might benefit from a deeper inspection for technical correctness.
> >
> > The reviewer has raised the score.

---

> > > ### Author Response · Authors · 2022-11-30
> > > **Thank you for the score increase; Regarding outstanding concerns**
> > >
> > > Thank you for taking the time to check our additional experiments in Safe AI Gym and raising your score. Regarding your outstanding concerns:
> > >
> > > > This simplest case would allow sequential combination of tasks where only one task is active, but the description in the paper appeared to hint to a more general composition. The reviewer does not suggest that linear combination is not useful, but rather that the limitation of what kinds of compositions can be captured (and what cannot be captured) by linear combination needs to be discussed. The concern is that the linear combination is going to be very restrictive.
> > >
> > > * We agree that making the rewards per RM node be linear (weighted) combinations instead of arbitrary real values is indeed a restriction.
> > > * We hope that the motivation for that restriction is clear, as it enables us to give guarantees about zero-shot composition.
> > > * While it is restricting, it is hopefully clear that it is still very general as it supports binary rewards in its simplest case, and also allows for the expression of linear preferences over any Boolean expression at each RM node.
> > > * We are happy to add a discussion on this in the paper using the example in Fig 1.  For example, compositions like $0.5 R_{coffee \wedge \neg decoration} + 0.5 R_{mail \wedge \neg decoration}$ and $[R_{coffee \wedge \neg decoration} ~ R_{mail \wedge \neg decoration}] \cdot softmax([R_{coffee \wedge \neg decoration} ~ R_{mail \wedge \neg decoration}])$ are supported, but not multiplicative ones like $(R_{coffee \wedge \neg decoration})^{0.5} (R_{mail \wedge \neg decoration})^{0.5}$. We believe our framework can be extended to support other types of compositions (like the multiplicative one) in future works.
> > >
> > > > No, LTL cannot be represented by a finite state automata similar to REs. There is some serious misunderstanding here. Buchi automata accept/reject infinite strings. The paper might benefit from a deeper inspection for technical correctness.
> > >
> > > * Thanks for the continued discussion on this point. We would like to begin by emphasizing that this paper focuses on tasks specified by reward machines. Prior works have already established the fact that reward machines can also be constructed from LTL [1] or regular expressions in general [2].
> > > * Please note that a Buchi automaton is a finite state automaton (at least in its common definition). See [3] (Chpt 5.12, Pg 83)
> > > * Buchi automata indeed accept/reject infinite strings, but not any type of infinite string. They specifically accept strings written in any omega-regular language (i.e they cannot accept/reject strings that require memory to determine acceptance/rejection). See [3] (Part II, Pg 39)
> > >
> > > [1] Camacho, Alberto, et al. "LTL and Beyond: Formal Languages for Reward Function Specification in Reinforcement Learning". IJCAI, 2019.
> > >
> > > [2] Icarte, Rodrigo Toro, et al. "Reward machines: Exploiting reward function structure in reinforcement learning". Journal of Artificial Intelligence Research (JAIR), 2022.
> > >
> > > [2] Rich, Elaine. "Automata, computability and complexity: theory and applications". Upper Saddle River: Pearson Prentice Hall, 2008.

---

> > > > ### Comment · Reviewer_MuPq · 2022-11-30
> > > > **Useful references for formal methods**
> > > >
> > > > > We believe our framework can be extended to support other types of compositions (like the multiplicative one) in future works.
> > > >
> > > > The reviewer does not see discussion in the paper for this optimism, and will suggest avoiding speculative claims in the paper.
> > > >
> > > > As mentioned in the comment, LTL cannot be represented by * a finite state automata similar to REs * . It is not similar to RE in the acceptance condition. It would be useful to look at a textbook on formal methods when creating the final draft. Examples of textbooks that could be useful: Logic in Computer Science: Modelling and Reasoning about Systems by Huth and Ryan, Handbook of Model Checking by Clarke et al.
> > > >
> > > > But having raised the score to positive, the reviewer continues to see value in the paper's core idea and is reasonably certain that a revision of the presentation to ensure accurate technical statements will make the paper stronger. The rebuttals have further assured the reviewer that the issue is just of presentation.

---

> > > > > ### Author Response · Authors · 2022-11-30
> > > > > **Thanks**
> > > > >
> > > > > We would like to thank the reviewer for engaging in the discussion and for the useful references.

---

### Official Review · Reviewer_38dZ · 2022-10-24

**Confidence:** 3
**Correctness:** 3
**Technical Novelty And Significance:** 2
**Empirical Novelty And Significance:** 2
**Recommendation:** 5

**Clarity, Quality, Novelty And Reproducibility:**

I should start by saying that I completely agree with the authors' motivation in developing compositional RL methods, and I think that most in the community would agree with this motivation---whether the compositionality should be built in by design as in this setup or learned automatically from data leads to broader disagreement, but still developing explicitly compositional techniques has good merit.

Perhaps my greatest concern with this paper is that I had a very difficult time finding details of the exact problem setting, approach, and experimental design. Here are some questions that may help clarify some things, which the authors hopefully can address and revise their manuscript to reflect the answers to these questions.
- The manuscript mentions multiple times that the proposed approach exhibits "concurrent compositionality". My understanding is that this stems directly from the use of Nange Tasse et al.'s (2020) [NT20 hereafter] method as the underlying mechanism for learning the primtive skills. Is this understanding correct? I believe this is correct based on this sentence form the beginning of Section 3.2: "We can leverage prior work to solve each individual task using a set of base primitive skills (NT20)." If this understanding is correct, I would encourage the authors to more explicltly state this (e.g., "We leverage NT20's method for learning the base primitive skills, which endows our method with the capability to achieve concurrent compositionality.") If not, what part of the proposed approach deals with the learning of the base skills in such a way?
	- As a side note, I would like to see a clearer discussion on the distinction between concurrent and temporal compositionality. If I understood correctly, the authors use concurrent compositionality to refer to goals specified as "achieve A and B", and temporal compositionality for goals such as "achieve A then B". If this is correct, in many cases the goals specified as concurrent might still be achieved sequentially (e.g., open door 1 and door 2---the agent presumably can't open both at the same time). Is the primary difference that the temporal compositionality explcitly encodes the temporal nature in the task specification?
- The manuscript also states that the primitive skills are learned in a reward-free setting. What exactly does this mean? I initially took it to mean that the task primitives were not manually specified, and the agent explored the environment in some exploratory way, say via curiosity-driven exploration. However, later parts of the manuscript seem to suggest that the task primitives are indeed manually specified. For example, in Section 4.3: "We first train the agent on three base task primitives: pick up blue objects, pick up purple objects, and pick up squares." Could the authors please clarify?
- I also have several questions about the experimental setting:
	- Sec. 4.1: What is the exact experimental setting? The primitive skills are trained on how many tasks and for how long? My interpretation of this sentence: "previous work (NT20) has shown that learning the WVFs takes longer than learning task-specific skills", is that these primitive skills are expensive to obtain. It would be useful to get a sense of how expensive it is for this domain.
	- Sec. 4.1: How is the agent achieving zero-shot performance on the single-task scenario (the rewards are high at the very beginning of the training curves) if, as the authors state, "In the single task domains, ... CRM, HRM and skill machines now cannot leverage prior knowledge"
	- Sec. 4.2: Could the authors for example compute the length of trajectories as a proxy for the qualitative results in Appendix 7? This way, we might get a quantitative comparison that better differentiates SM and QL-SM
	- Sec. 4.2: "skill machines with Q-learning always begin with a significantly higher reward" -- there is no discernible difference in reward w.r.t. SM in Fig 3. What do the authors mean with this claim?
	- Sec 4.3: "In such cases, learning new skills few-shot by leveraging the SM would guarantee convergence to optimal policies as demonstrated in Section 4.2" -- was this validated empirically for the image-based domain?

If my understanding is mostly correct, then the approach uses NT20 to learn primitive skills, and then uses value iteration over RMs to discover how to sequence the skills to achieve temporally specified goals (e.g., in LTL). If this is the case, the novel contribution of the paper in terms of an algorithm is described in these 2 sentences: "To achieve this, we first plan over the reward machine (using value iteration, for example) to obtain Q-values for each transition. We then select the skills for each SM state greedily." Hopefully a revised version of this manuscript can provide a lot more detail, to gauge more accurately the extent of the contributions. The theory in theorem 3 essentially says that there exists some SM that can solve any task in the given task space, but is this also the case for the SMs found via the approximate planning-based solution?


############## Additional feedback ##############

The following points are provided as feedback to hopefully help better shape the submitted manuscript, but did not impact my recommendation in a major way.

Intro
- So far I'm pretty excited to see where this goes. The problem seems interesting and the promises of the results are high.

Sec 3
- It took 4 full pages to cover the background and definitions. While these are not super commonly known results, and so it might make sense to go over them in some detail, maybe it would be better to compress these sections a little to give more room for technical details about the proposed solutions.
- The running example domain is a nice way of guiding the reader

Sec 3.1
- Is the set of task-specific goal rewards for the task primitives manually specified? If so, how? And how does this relate to the notion of "reward-free" specified earlier in the paper for learning the base skills?
- I didn't follow how Lemma 2 in NT20 implies the proof in Theorem 1 here. Why can we assume that pi* is the same? For self-containedness, it might be worth to clarify that more explitly in the Appendix.
- Why do the skills for the task primitives enable solving logical composition? Are we assuming that we are directly following NT20 and so the problem of logical composition is solved via their method?

Sec 3.2
- It seems, up to this point, that these preference functions would be completely unlearnable
- It also seems that, for an exact solution, a different one would have to be learned for each new RM (= task)

Sec 3.3
- My understanding is that the approximate planning over RMs chooses one-hot preferences, as shown in the theorem

Sec 5
- Isn't the ability to handle negation (and all forms of concurrent composition) just a direct consequence of using NT20 under the hood?
- The comparison against options seems a bit unfair because SMs directly use planning to achieve long-term reasoning. Why not compare against similar planning-based approaches?

Typos/style/grammar/layout
- Multi-task or multitask? I saw both in different parts of the paper
- Sec 4.2: there are no orange curves in Fig 3

Supplement
- Source code is not included with the submission, which could have helped clarify some of the details of the proposed solution.

**Strength And Weaknesses:**

########### Strengths ###########
- The motivation for the work is excellent: using RL to train primitive skills that can serve as a basis for solving a multitude of tasks via composition, both concurrently and temporally, as specified by RMs

########### Weaknesses ###########
- I had a difficult time following the details of the approach and the corresponding theory behind it. Perhaps the authors could add more technical details to their manuscript?
- I also had difficulty understanding the exact experimental setting used throughout Section 4
- If my understanding was correct, then I am uncertain about the novelty of the contributions in this work. I hope the authors can clarify this during the discussion period.


**Summary Of The Paper:**

The submission introduces skill machines (SM), an approach to compositionally generalize over a space of tasks defined as reward machines (RM). In particular, this formulation enables compositional generalization both over concurrent goals (e.g., achieve a ^ b) and temporal goals (e.g., achieve a "then" b). The proposed approach uses planning over the RM leveraging the known RM transition function, and then chains pre-trained skills together to follow the plan. Theoretically, the authors show that SMs are sound, in that if a solution to the task exists in terms of the primitive skills, the SM will find it. Empirically, the method achieves zero-shot generalization, and improves with further training, on one set of grid world tasks and one set of image-based tasks.


**Summary Of The Review:**

Overall, I think the motivation for this work is strong, but had a hard time finding the technical details needed to precisely understand what the technical contributions of the submission are. I strongly encourage the authors to provide as much detail as possible about their approach in a revised manuscript and to engage in the discussion so I can better understand their contributions. In the meantime, with the manuscript in its current state, I cannot recommend it for acceptance. I will use a low confidence score for my rating to indicate that I have a number of questions that, if clarified, might significantly alter my assessment of the work.

############# Update after rebuttal ###############

I am increasing my score from 3 (reject) to 5 (marginally below threshold) per the discussion with the authors, and increasing my confidence accordingly.

---

> ### Author Response · Authors · 2022-11-19
> **Thanks for the careful review. We have updated the paper to clarify your concerns. [1/2]**
>
> Thanks for your thorough review of our paper. Also thanks to your clarity concerns, we have updated the paper to clarify our notations and the definitions, specifically that of tasks (Def 3), and primitives (Def 4). We believe the paper is much clearer now and hope it addresses most of your concerns.
> Our main revisions are color-coded. Violet for your concerns, and red for yours and other reviewers.
> We hope to have also addressed your specific points below:
>
> > If my understanding was correct, then I am uncertain about the novelty of the contributions in this work. I hope the authors can clarify this during the discussion period.
>    * This is a summary of our main contributions:
>    * We propose a definition of skill primitives that enables them to be composable logically (according to guarantees from NT20), linearly (theorem 1) and temporally zero-shot.
>    * Given a task specified by an RM, we propose a definition of skill machines that encode how to compose skill primitives to solve the task zero-shot.
>    * Given a task specified using a simple RM, we show how to learn a skill machine from the simple RM such that it is guaranteed to be satisficing (it is guaranteed to solve the task).
>    * We propose a method with convergence guarantees for learning new skills few-shot by leveraging the zero-shot SMs.
>    * We demonstrate our approach empirically with several tasks in both tabular and high dimensional function approximation domains. Our results show that the skill primitives learned in an environment (according to our definition of task primitives) can be successfully composed both logically, linearly, and temporally in skill machines to solve tasks zero-shot and few-shot, significantly outperforming all baselines.
>
> > I had a difficult time following the details of the approach and the corresponding theory behind it. Perhaps the authors could add more technical details to their manuscript?
>    * We have clarified the technical details of the approach (see the color-coded regions)
>
> > I also had difficulty understanding the exact experimental setting used throughout Section 4
>    * We have added more descriptions of the experimental setting at the beginning of Sec 4 and full details in the Appendix.
>
> > The manuscript mentions multiple times that the proposed approach exhibits "concurrent compositionality". My understanding is that this stems directly from the use of Nange Tasse et al.'s (2020) [NT20 hereafter] method as the underlying mechanism for learning the primtive skills. Is this understanding correct?
>    * Yes, but not directly. Since NT20 does not tackle temporal logic composition, it is not clear how to leverage that framework in a way that achieves zero-shot logic and temporal compositions, with guarantees.
>    * We show how to define task and skill primitives in a way that is compatible with both their framework and reward machines, enabling us to leverage their zero-shot logic composition results to achieve zero-shot temporal logic. This is a core contribution of this work.
>
> > As a side note, I would like to see a clearer discussion on the distinction between concurrent and temporal compositionality. Is the primary difference that the temporal compositionality explcitly encodes the temporal nature in the task specification?
>    * Yes. The door example you gave is correct, and is similar to the example in Figure 1 “deliver coffee and mail to the office without breaking any decoration”.
>    * Another example is "pickup a blue object then a box" is temporal composition of “blue” and “box”, while pickup “pickup a blue box” is a concurrent composition of “blue” and “box”.
>    * We have added this example to the introduction, and have also changed “concurrent” to “spatial” for better clarity.
>
> > The manuscript also states that the primitive skills are learned in a reward-free setting. What exactly does this mean? I initially took it to mean that the task primitives were not manually specified, and the agent explored the environment in some exploratory way, say via curiosity-driven exploration. However, later parts of the manuscript seem to suggest that the task primitives are indeed manually specified. For example, in Section 4.3: "We first train the agent on three base task primitives: pick up blue objects, pick up purple objects, and pick up squares." Could the authors please clarify?
>    * By reward-free we mean in the absence of task rewards from a reward machine. We have added this to the text for more clarity.
>    * Given a background MDP and a labeling function with propositions P, the agent learns a skill primitive $Q_p$ (a WVF) for a task primitive $M_p$ using the rewards and transitions as defined in Def 4. For our experiments, we learn |P| base skill primitives, each of which corresponds to achieving each of the propositions in P (they are base primitives because they can then be composed to obtain any of the other $2^{2^P}$ primitives). We have clarified this in the text.

---

> > ### Author Response · Authors · 2022-11-19
> > **Response Continuation [2/2]**
> >
> >
> > > Sec. 4.1: What is the exact experimental setting? The primitive skills are trained on how many tasks and for how long? My interpretation of this sentence: "previous work (NT20) has shown that learning the WVFs takes longer than learning task-specific skills", is that these primitive skills are expensive to obtain. It would be useful to get a sense of how expensive it is for this domain.
> >    * We have added more descriptions of the experimental setting at the beginning of Sec 4 and full details in the Appendix.
> >    * The skill primitives are trained for 200000 time steps, the same as the runs for Fig 3.
> >    * We mention that WVFs takes longer than learning task-specific skills (NT20 shows the difference scales by a constant) to highlight that we don’t train them to optimality. We used  200000 time steps because that is the amount of time it took for at least one of the baselines to converge.
> >    * If we are only interested in a single task (or even a small number of tasks) with small reward machines, then using skill machines is probably not a good tradeoff since the WVFs may take longer to train. However, in settings where we are interested in multiple (possibly infinite) number of tasks, with RMs which can be arbitrarily large and complex (meaning long complex temporal horizons, like task 4 Fig 3.b), then using skill machines to achieve zero-shot logical and temporal composition becomes necessary even if the WVF of skill primitives take some time to learn.
> >
> > > Sec. 4.1: How is the agent achieving zero-shot performance on the single-task scenario (the rewards are high at the very beginning of the training curves) if, as the authors state, "In the single task domains, ... CRM, HRM and skill machines now cannot leverage prior knowledge"
> >    * We hope this is clearer in the paper now. As mentioned above, we pretrain the base skill primitives in the environment (without reward machines). The skill machine then composes these primitives appropriately to have zero-shot near-optimal performance from the start when given tasks to solve.
> >
> > > Sec. 4.2: Could the authors for example compute the length of trajectories as a proxy for the qualitative results in Appendix 7?
> >    * This is equivalent to what Fig 3 shows, since the rewards are 1 for successful completion of the tasks and 0 everywhere else, and Fig 3 shows the number of rewards obtained in every 5000 steps. We computed the length per episode and it gave similar graphs.
> >    * The difference in trajectory lengths are only slightly visible because of the standard deviations from using random start positions (the zero-shot results are optimal in most start positions). See Fig 3.a.
> >
> > > Sec. 4.2: "skill machines with Q-learning always begin with a significantly higher reward" -- there is no discernible difference in reward w.r.t. SM in Fig 3. What do the authors mean with this claim?
> >    * We mention this, “skill machines with Q-learning always begin with a significantly higher reward and converge on their final performance faster than all benchmarks—except the zero-shot one which is (near) optimal in all cases”.
> >    * We are happy to rephrase it to be clearer
> >
> > > Sec 4.3: "In such cases, learning new skills few-shot by leveraging the SM would guarantee convergence to optimal policies as demonstrated in Section 4.2" -- was this validated empirically for the image-based domain?
> >    * We meant here that learning new skills fewshot using tabular Q-learning (as was done in the office world) would guarantee convergence in the limit. Unfortunately, this is clearly impractical (since there is about ~$10^10$ environment states), and using function approximation as we did removes any convergence guarantee (we saw no improvements when we tried it).
> >    * We realise that the phrasing there was a bit unclear, and have appropriately revised it.
> >
> > > The theory in theorem 3 essentially says that there exists some SM that can solve any task in the given task space, but is this also the case for the SMs found via the approximate planning-based solution?
> >    * Yes, this is the exact statement of Theorem 3.
> >    * To elaborate more, Theorem 3 says that the SMs found via the planning-based solution solves any tasks defined by simple reward machines. We meant the resulting SM is approximate because it is satisficing but not necessarily optimal.
> >
> > > Sec 5. The comparison against options seems a bit unfair because SMs directly use planning to achieve long-term reasoning. Why not compare against similar planning-based approaches?
> >    * All the baselines (QL, CRM, HRM) also have planning variants (QL-RS, CRM-RS, HRM-RS) which we compared against. They use the values obtained from value iteration over the reward machines for reward shaping (Icarte et al., 2018).

---

> > > ### Comment · Reviewer_38dZ · 2022-11-21
> > > **Response to authors**
> > >
> > > Given the authors' reply, I do believe I have gained a better understanding of their contributions. In particular, the main contribution appears to be the definition of skill machines in a way that enables 1) the use of primitive skills that are already concurrently (now spatially) compositional, and 2) a simple planning-based algorithm for combining these primitive skills temporally. While this became clear through a combination of the authors' response, reading the other reviews, and parts of the authors' revisions, it is my perception that the manuscript by itself is still a bit hard to follow.
> > > - For example, I found the red paragraph towards the center of page 5 to be most informative for understanding the "skill primitives." Yet, the phrasing is still quite convoluted, using "skill primitives" as any composition (presumably concurrent/spatial per NT20) of "base skill primitives", without really explicitly stating this in a clear enough way---I am only somewhat sure that my understanding of this paragraph is correct, and only from combining my previous understanding of NT20 with the authors' responses and other reviews. I generally think the use of "base" and "primitive" as distinct concepts is bound to create confusion.
> > > - As another example, reviewer UCg6 raised the concern that Section 3.3, which describes the method for learning to temporally combine primitive skills via value iteration, was also unclear and failed to precisely describe how the preferences over actions were learned. I believe that this continues to be unclear in the revised manuscript.
> > > My sense is that the paper is similarly difficult to follow throughout, despite the authors' efforts in revising it.
> > >
> > > Regarding the experimental setting, I now understand that, even in the single-task setting, the authors' proposed method is still pre-training the skill primitives, which enables the observed zero-shot performance. However, my other concerns still remain: the results don't validate the use of Q-learning to fine-tune the zero-shot policy. Only the results in Figure 9 in the appendix (qualitatively) distinguish SM and QL-SM, and given the fact that the rewards in Figure 3 are equivalent, it is difficult to say if these qualitative results are informative. Results on the image-based setting would likely have been more interesting for distinguishing these two, but this was not empirically validated.
> > >
> > > That being said, my renewed understanding of the contributions does lead me to increase my perceived novelty of the submission, and I therefore will increase my rating to weak reject.

---

> > > > ### Author Response · Authors · 2022-11-24
> > > > **Reply to Reviewer**
> > > >
> > > > Thanks for checking out our revisions and updating your score. We are grateful that our revisions have given you a better understanding of our work. Regarding your outstanding concerns,
> > > >
> > > >
> > > > >  I generally think the use of "base" and "primitive" as distinct concepts is bound to create confusion.
> > > > * That is a good point. By "base" primitives we mean the smallest subset of primitives that can be spatially composed to obtain any other primitive. We are happy to call it **basis** primitives instead for better clarity.
> > > >
> > > > > As another example, reviewer UCg6 raised the concern that Section 3.3, which describes the method for learning to temporally combine primitive skills via value iteration, was also unclear and failed to precisely describe how the preferences over actions were learned.
> > > > * As you correctly mentioned in your original review, "the approximate planning over RMs chooses one-hot preferences, as shown in the theorem". This is also illustrated in Figure 2.
> > > > * We also clarified it to reviewer UCg6.
> > > > * We are happy to clarify any outstanding concerns about this.
> > > >
> > > > > Regarding the experimental setting, I now understand that, even in the single-task setting, the authors' proposed method is still pre-training the skill primitives, which enables the observed zero-shot performance. However, my other concerns still remain: the results don't validate the use of Q-learning to fine-tune the zero-shot policy.
> > > >
> > > > * We understand that the experiments in the office gridworld do not quantitatively demonstrate the benefit of our few-shot learning compared to our zero-shot composition. The point here was to show that the few-shot agent can start with the same performance as the zero-shot agent, and improve over that using Q-learning, as demonstrated qualitatively by the trajectories in Fig 9. Note that these trajectories are the actual trajectories followed by the zero-shot and few-shot agents starting from the shown start position. We are happy to update the captions with the length of those trajectories:
> > > >   * **Task 1 zero-shot $\to$ 39 steps | Task 1 few-shot $\to$ 19 steps; Task 2 zero-shot $\to$ 45 steps | Task 2 few-shot $\to$ 37 steps; Task 3 zero-shot $\to$ 53 steps | Task 3 few-shot $\to$ 53 steps.**
> > > >
> > > > * We also ran the same single-task experiments for task 1 in Table 1 (the simplest, least interesting task), and did observe a significant quantitative difference between the zero-shot and few-shot agents. We are happy to update the paper with it.
> > > >
> > > > > Results on the image-based setting would likely have been more interesting for distinguishing these two, but this was not empirically validated.
> > > > * For both the image-based setting (Fig 4) and continuous setting (Fig 5), we observed no quantitative nor qualitative difference in performance between the zero-shot and few-shot agents.
> > > >
> > > >
> > > > Finally, we would like to emphasize that the **zero-shot temporal logic composition** is our main contribution, which we have provided **theoretical guarantees** for, demonstrated empirically that it **drastically outperforms all the prior state-of-the-art methods [1]**, and demonstrated that it works in **tabular, image-based, and continuous settings**.  We are happy to discuss any further concerns you may have.
> > > >
> > > > [1] Icarte, Rodrigo Toro, et al. "Reward machines: Exploiting reward function structure in reinforcement learning." Journal of Artificial Intelligence Research; 2022.

---

> > > > > ### Author Response · Authors · 2022-12-08
> > > > > **Following up on the reviewer's concerns**
> > > > >
> > > > > Since the discussion period is coming to an end, we would like to clarify whether the above response addresses the reviewer's outstanding concerns. Additionally, if we can clarify any further concerns we would be glad to do so.
> > > > >
> > > > > We thank the reviewer for their engagement during the discussion and their help in improving our paper.

---

### Official Review · Reviewer_ronJ · 2022-10-24

**Confidence:** 3
**Clarity, Quality, Novelty And Reproducibility:** The proposed method is novel. This pa…
**Correctness:** 3
**Technical Novelty And Significance:** 3
**Empirical Novelty And Significance:** 3
**Recommendation:** 6

**Strength And Weaknesses:**

Strength:
The proposed method takes use of the finite state machine to solve complex RL tasks. The proposed method guarantees that the resulting policy adheres to the logical task specification. The idea is promising given the fact that the algorithm can learn policies for multiple tasks at the same time. Sufficient experiments are presented to support the statement in this paper.


Weakness:
Since the proposed method is a finite state machine based, I expect the algorithm suffers from the curse of the dimensionality. The example of office gridworld is simple and small in terms of the goal, state, and action spaces. I would like to read a bit more about how to generalize the proposed method in the case when the spaces are much larger which is more likely to be true in real applications.
The number of goals is very limited in the experiments presented here. It will be great if the authors can present some examples that has larger goal space.

**Summary Of The Paper:**

This paper considered a multi-task reinforcement learning problem where all tasks share the same state space, action spaces, and the transition dynamics. A skill machine is proposed to solve complex task involving temporal and concurrent composition which can be learned from reward machine. The authors prove that an agent can solve it while adhering to any constraints. Through experiments, the authors demonstrate the efficiency of the proposed method in several environments. The result shows that the proposed method can provide a near-optimal behavior for a long horizon task without further learning.

**Summary Of The Review:**

The multitask reinforcement learning problem is important. But I have concern about the scalability of the proposed method to a more complex real problems.

---

> ### Author Response · Authors · 2022-11-17
> **Thanks for the careful review. We have updated the paper to clarify your concerns.**
>
> Thanks for your careful review of our paper. We hope to have addressed your concerns below:
>
> > Since the proposed method is a finite state machine based, I expect the algorithm suffers from the curse of the dimensionality.
>    * Our work addresses this exact problem that was faced by previous works.
>    * In this setting where agents need to solve several temporal logic tasks specified by reward machines/finite state machines (or LTL), the number of possible tasks is infinite and the size of the RMs can be combinatorially huge (meaning extremely complex and long temporal horizons).
>    * Our work enables an agent to learn skill primitives in an environment (in the absence of any reward machine), then compose them zero-shot to immediately solve any solvable RM without further learning in the environment.
>    * Thanks for this point. We have added a discussion about this in the updated paper (page 5), and we believe it makes the paper clearer.
>
> > I would like to read a bit more about how to generalize the proposed method in the case when the spaces (action, state, goal) are much larger which is more likely to be true in real applications.
>    * That is a good point and is always a concern in the choice of domains used.
>    * Please note that we used similar domains to the standard domains used in previous related works [1,2,3] where an agent needs to navigate to objects, since they are representative and still very hard. Similarly to those works, our approach is agnostic to the specific environment, i.e. any suitable RL algorithm can be used to learn the value functions for our primitives (in the defined primitive MDPs, Def 4). Hence how well our approach scales (in action/state/goal spaces) is directly proportional to how well the chosen standard RL algorithm scales.
>    * For example, our function approximation experiments (the moving targets domain) have 84x84x3 RGB pixel observations, and we simply use a DQN to learn the skill primitives (using the network architecture from the DQN nature paper).
>
>
> > The number of goals is very limited in the experiments presented here. It will be great if the authors can present some examples that has larger goal space.
>    * The moving targets domain has 6 different objects, which is the same number as previous works [1,2]. This gives $2^6$ Boolean expressions/goals (sets of picked objects, e.g {blue square, purple circle}) that the reward machine transitions can be defined over. This also gives $2^{2^6}  \approx 10^{19}$ possible sets of goals the agent may need to reach for each RM state (e.g pickup {blue square, purple circle} or {beige square, purple circle} or {beige circle, beige square})
>    * Note also that the agent and object positions are randomly placed in the environment, making the state space massive (~$10^{10}$).
>
>
> [1] Geraud Nangue Tasse, Steven James, and Benjamin Rosman. A Boolean task algebra for reinforcement learning.
>
> [2] Rodrigo Toro Icarte, Toryn Klassen, Richard Valenzano, and Sheila McIlraith. Using reward machines for high-level task specification and decomposition in reinforcement learning.
>
> [3] André Barreto, Shaobo Hou, Diana Borsa, David Silver, and Doina Precup. Fast reinforcement learning with generalized policy updates

---

### Official Review · Reviewer_UCg6 · 2022-10-25

**Confidence:** 4
**Correctness:** 3
**Technical Novelty And Significance:** 3
**Empirical Novelty And Significance:** 3
**Recommendation:** 6

**Clarity, Quality, Novelty And Reproducibility:**

The clarity (and with that the reproducibility) is not adequate yet, I think. The novelty of the work is clear; I have not seen any combination of these two approaches yet, and it makes a lot of sense. The quality of the work (technically) is good enough, I think, but parts of it are not well described which make it not possible (for me) to fully evaluate it properly. The description and motivation of the technical parts needs work.

**Strength And Weaknesses:**

The main strength of the paper is the proposed combination of methods. I think it makes a lot of sense, it is based on very recent work, and I see the potential. The paper appears polished, sketches related work, and the experiments show progress.

A main weakness is the description of the method, the buildup of the story and the motivations/examples. This paper is in my top-k papers ever that are hard to review, hence any remark I make here could be the result of a misunderstanding on my side (disclaimer) but I do think it is mainly due to a poor description. I'm very knowledgeable of all the related work, and in particular the temporal/logical side of the matters, but I have a hard time understanding the method from its particular description in the text (and the appendices do not help much, since not much additional information is included).

The introduction section and Section 2 up to 2.2. are very clear in terms of setup, contributions and approach. Section 2.1. just gives the flavor of multi-task learning, compositions of Q-functions and so on, that are necessary for the rest of the paper. However, Section 2.2. is (in my opinion, and I re-checked the original papers on this topic) not understandable for the general (RL) reader, since it involves many things in only three small paragraphs, even though it implicitly assumes knowledge of how such machines work. For example, the "joint MDP" is used, but this relies on knowing more about temporal logic, the ability to compile into automata and the cross-product of machines with the MDP to create such a joint MDP. Later on in the paper, additional mentionings are made of for example "product" (page 4) and "cross product" (page 5) but never really explained. At this stage it would or could be an obvious thing to say that if we can see the problem as a joint MDP, and I mean the full product model, it again is a Markovian problem (which is the trick in many other papers in this area), hence the approach in Section 2.1. would already apply immediately (since the temporal dimension is compiled out essentially), and the combination will work. Ofcourse... you want to keep as much of the method on an abstract level (hence the reward machine) and you need more. In Section 3.1. the task space is first again formalized as a full product model and the compositional mechanism of Section 2.1. generalized to linear preferences over task primitives. I think this all makes sense, and is clear. The only thing that is not helping, is all the subtle differences of tasks, primitives, skills, goals, skill primitives and many other terms; in my opinion this could be made more clear throughout the paper too.

Section 3.3 is where I think most of the confusion comes from. I think that the motivation and definition of skill machines needs much more explanation and formalization. I you look at the Figures 1 and 2, we see that they are essentially reward machines (in structure) but the difference lies in the "preferences" on the lower parts of the nodes, which give preferences over action choices (in the reward machines, thus not actually action choices, but the condition of going from one high-level node to another). But, even though I have some general understanding of how this would be done (based on earlier definitions in the paper) the text and formalization as given here, are incomplete to me. The motivations for the preferences (state transitions and goals) and how they can be found, and what they mean, does not become clear enough (also not from the proofs of the definitions). Also the additional description of additional learning steps (on page 6) to finetune approximate SMs, is not clear enough for me.

Section 4 on the experiments has mixed quality, in my opinion. I think the experimental setup and selection of experiments is fine, and clearly is targeted at showing the composition of temporal skills and the reuse or zero-shot computation of new skills is clear. The last experiment with the moving domain, and the exact (motivation for) the particular setup in learning needs more description. But overall, the texts in this section are somehow describing a lot, yet at the same time not very clear and structured. When only looking at the graphs, I see the main message, but the texts do not help much. Overall, I see that what the authors want to show is in the results: for example, the composition of "pick up blue objects or squares, but never blue squares" and then repeating it, is a clear example of the composition of temporal skills, and it works.

Additional aspects:
- Just before Section 2: "high-dimensional video game". This does seem relevant for the experiments at all, right? Same for the remarks about it in Section 4.3. itself.
- Even though I think Section 2.1. contains enough information, I do wish more information would be given about the limitations, and also on the exact motivation for the third rule on negation (generally it looks like Fuzzy operaters by the way, but that is not relevant here).
- Section 2.2. talks about general and simple RMs but it is sometimes unclear which one is the main one to go later on. Maybe be more clear about it here. This section could also use a big illustration of things. Also not clear whether the "function" or "scalar" difference is intuitive without illustrations.
- It is not explained how to move through a (*) location without "breaking any decoration" (see Fig 1 left vs caption).
- The formalization in Def 2. uses subscript 0 essentially for the background MPD even though in this context (with automata) it can be confused with the starting state.
- The discussion of multi-task setups, CRM, HRM and the use of experience for multiple skills is something that is known in hierarchical RL as well, and options. This aspect could be discussed more in detail here, combined with the last part of Section 3 on additional learning phases in the SMs.
- I think the first line of the conclusion is confusing (in the light of the remarks above too) in the sense that skills machines are "finite state machines that can be learned...". It is factually not true, since the automata themselves are not learned (but some parameters are computed or finetuned) and maybe this illustrates some of the confusion I have with parts of the paper still.


**Summary Of The Paper:**

This paper is a combination of two previous works. One is about logical (boolean) compositional mechanisms for multitask learning, and the other is about logical formalisms for temporal skills, both in the context of typical reinforcement learning problems. The idea is to lift the first (composing behaviors in a structured way from individual Q-functions basically) to the case of of temporally extended behaviors, in this case modeled as reward machines. The overall idea is to come up with so-called skill machines, which can represent both types of structure. The idea is evaluated on a couple of domains, where the goal is to show that many different (composed) skills can be learned (or zero-shot, computed/assembled) from a couple of basic (temporal) skills.

**Summary Of The Review:**

A mixed evaluation, and one of these harder to review papers in my opinion. I clearly see the technical and conceptual positives in this paper, and I see the potential and the interesting outcomes of the paper. Yet, the description of some parts of the technical aspects is not yet clear enough for me to fully evaluate it. I do not want to kill a (potentially) good paper, hence I vote for a (weak) accept and await the discussion to see whether it can be improved still.

---

> ### Author Response · Authors · 2022-11-17
> **Thanks for the thorough review. We have updated the paper to address your concerns.**
>
>
> Thanks for your thorough review of our paper. Also thanks to your clarity concerns, we have updated the paper to clarify our notations and the definitions, specifically that of tasks (Def 3), and primitives (Def 4). We believe the paper is much clearer now and hope it addresses most of your concerns.
>
> Our main revisions specific to only your concerns have been color-coded in blue and the rest in red (they are mostly in red since they are also relevant to the other reviewers).
>
> We hope to have also addressed your specific points below:
>
> > A main weakness is the description of the method, the buildup of the story and the motivations/examples.
>    * We have moved the running example (Fig 1) to the introduction and used it to better describe and motivate the various main points in our approach.
>
> > "product" (page 4) and "cross product" (page 5) but never really explained.
>    * Thanks for this suggestion. We have clarified the meaning of the product between the background MDP and an RM, and added its specific definition (Product MDPRM, Def 2)
>
> > The only thing that is not helping, is all the subtle differences of tasks, primitives, skills, goals, skill primitives and many other terms.
>    * We have clarified the specific definition of temporal logic tasks (Def 3), task primitives, and skill primitives (Def 4), and have added clarifications to other terms.
>    * Temporal logic tasks (Def 3) are product MDPRMs whose rewards per RM states are linear preferences over Boolean expressions. Fig 1 shows an example.
>    * Task primitives (Def 4) correspond to temporally atomic tasks where the agent needs to achieve a set of propositions. For example “pickup coffee” ( which is F coffee in LTL). To leverage the Boolean composition results of Tasse et al, the task primitives are defined as goal based MDPs, where goals are simply sets of propositions. Skill primitives are then defined simply as the value functions of task primitives.
>
> > Section 3.3 is where I think most of the confusion comes from. I think that the motivation and definition of skill machines needs much more explanation and formalization.
>    * We hope this has been clarified given the clarified tasks and primitives.
>    * We have made the definition of skill machines more specific given the clarified definitions of tasks and primitives.
>
> > The motivations for the preferences (state transitions and goals) and how they can be found, and what they mean, does not become clear enough
>    * The preference over transitions $\mathbf{w}_U$ represents cases where there is not necessarily a single desirable transition to follow given the current SM state. This is illustrated by the SM in Fig 1b, where mail and coffee are equally desirable at the initial state. Similarly, the preference over goals $\mathbf{w}_G$ represents cases where there may be a single desirable task, but its goals are not necessarily equally desirable given the environment state---for example when the agent needs to first pick up coffee but there are two coffee locations.
>    * Theorem 3 says we can find $\mathbf{w}_U$ by planning over a simple RM and picking the best transitions, and $\mathbf{w}_G$ is simply chosen to be the best goal using the composed WVFs of skill primitives.
>
> > Section 4 on the experiments has mixed quality, in my opinion. … But overall, the texts in this section are somehow describing a lot, yet at the same time not very clear and structured.
>    * We have focussed the explanations and clarified the experiment setup.
>
> > Just before Section 2: "high-dimensional video game". This does seem relevant for the experiments at all, right? Same for the remarks about it in Section 4.3. Itself.
>    * It is relevant because our function approximation experiments (the moving targets domain) have 84x84x3 RGB pixel observations. This shows that our approach also works in settings where function approximation is required, since any suitable RL algorithm can be used to learn the skill primitives.
>
> > It is not explained how to move through a (*) location without "breaking any decoration" (see Fig 1 left vs caption).
>   * This can be done by using Q-learning over the product MDPRM, or it can be done zero-shot using the skill machine corresponding to that RM
>   * We have revised the Figure and caption to clarify this.
>
> > The first line of the conclusion is confusing in the sense that skills machines are "finite state machines that can be learned...". It is factually not true, since the automata themselves are not learned.
>    * We hope this makes sense now given our revisions.
>    * Given a task defined by an RM, the corresponding skill machine is parametrised by a preference over transitions $\mathbf{w}_U$ (which can be learned by planning over the RM, Fig 2), and a preference over goals $\mathbf{w}_G$  (which can be computed from learned skill primitives).
>    * We are happy to change the phrasing to “FSMs whose parameters can be learned from RMs” if that makes more sense.

---

> > ### Comment · Reviewer_UCg6 · 2022-11-23
> > **thanks**
> >
> > Thanks for the elaborate reply, and I'm glad my positive evaluation is now based on more evidence in the revised paper. Given all the improvements, and rest of the materials here, I'm confirming my "accept" for this paper. My feeling it would be something like approaching a "7", even though some of my concerns (mainly about clarity of several things) are still valid (although to a lesser extent now).

---

### Author Response · Authors · 2022-11-19
**Paper updated with clarity revisions and additional experiments in the requested continuous setting (Safety AI Gym)**

We would like to thank all the reviewers for their thorough review of our paper and constructive feedbacks.

The main concern was about the clarity of the paper, both in the theory and experiments.
* Thanks to that and their suggestions, we have updated the paper to clarify our notations and definitions, specifically that of tasks (Def 3), and primitives (Def 4).
* We have also clarified our experiments, and added the full details of their setup in Appendix A.4.
* Finally, as requested by Reviewers ronj and MuPq, we have added additional experiments in the Appendix A.3 demonstrating that our approach also works in hard continuous environments (using the Safety AI Gym domain suggested by reviewer MuPq)

We believe the paper is much clearer now and hope it addresses all of the reviewer's concerns.

Our main revisions specific to each reviewer have been color-coded and those relevant to all are in red.

---

### Decision · Program_Chairs · 2023-01-20

**Decision:**

Reject

**Justification For Why Not Higher Score:**

We all worked hard to understand the paper but were not satisfied with our understanding.  This means that other readers are likely to find it difficult to understand and to replicate.

**Justification For Why Not Lower Score:**

N/A

**Metareview: Summary, Strengths And Weaknesses:**

This paper addresses a very important problem, providing a way to learn a basis set of low-level skills and then combine them both in "series" (to satisfy temporal logic constraints) and "parallel" (to satisfy Boolean combinations of objectives).  To do so, it combines existing strategies for handing temporal-logic objectives via reward machines and for Boolean combination of goals.  These are new, interesting, complimentary mechanisms and it is not trivial to combine them, so this paper offers significant novelty.

All reviewers and the AC found the paper to be interesting and potentially important.  However, we all also found it very difficult to understand (and the reviewers were well qualified to be able to understand a paper in this area).   Even after significant improvements by the authors, the most critical (sections 3.2 and 3.3) remain elusive.  And, in the experiment descriptions, it was never clear exactly what was learned, when.  There is learning of primitives, analytic (zero-shot) composition of them given a goal spec, but then also something that happens "few-shot" and a mention of the possibility of fine-tuning in the conclusion.

To really succeed in communicating, it might be helpful to have two things:
1.  A data-flow diagram that explains the computational stages of the overall approach:  what is the input to the entire system?  how are the primitives learned?  what happens when a new task specification arises?  etc.
2.  A more intuitive description of the construction in section 3, and then step-by-step explanation of the example in figure 2.   (Even at the level of detail of explaining the relationship between definition 5 and the skill-machine diagrams.)

We recognize that papers have finite space;  it might be that a somewhat more abstract, intuitive, and compressed description of the two pre-existing composition techniques would allow more room for motivation, intuition and examples of the new contributions.

We would strongly encourage the authors to revise the paper for future submission and look forward to reading it.

**Summary Of Ac-Reviewer Meeting:**

The participants agreed that there was interesting content in the paper but in the end it was just not clear enough.